

# Impacts of Large-Scale Circulation on Urban Ambient Concentrations
# of Gaseous Elemental Mercury in New York, USA

Huiting Mao[1†], Dolly Hall[2], Zhuyun Ye[1], Ying Zhou[1], Dirk Felton[3], and Leiming Zhang[4]
[1]Department of Chemistry, State University of New York College of Environmental Science and
Forestry, Syracuse, NY 13210
[2]Department of Atmospheric and Oceanic Science, University of Maryland, College Park, MD
7    20742
[3]Bureau of Air Quality Surveillance, Division of Air Resources, New York State Department of
Environmental Conservation, Albany, NY 12233
[4]Air Quality Research Division, Science and Technology Branch, Environment and Climate
Change Canada, Toronto, M3H 5T4, Canada
[†]Correspondence to: hmao@esf.edu




## Abstract

The impact of large-scale circulation on urban gaseous elemental mercury (GEM) was
investigated through analysis of 2008 – 2015 measurement data from an urban site in New York
City (NYC), New York, USA.  Distinct annual cycles were observed in 2009-2010 with mixing
ratios in warm seasons (i.e. spring-summer) 10-20 ppqv (~10%-25%) higher than in cool seasons
(i.e. fall-winter).  This annual cycle was disrupted in 2011 by an anomalously strong influence of
the North American trough in that warm season and was reproduced in 2014 with annual
amplitude enhanced up to ~70 ppqv associated with a particularly strong Bermuda High.  North
American trough axis index (TAI) and intensity index (TII) were used to characterize the effect
of the North American trough on NYC GEM especially in winter and summer.  The intensity and
position of the Bermuda High had a significant impact on GEM in warm seasons supported by a
strong correlation ($r$ reaching 0.96, $p<0.05$) of GEM with Bermuda High intensity indices in
summer. Regional influence on NYC GEM was supported by the GEM-carbon monoxide (CO)
correlation with $r$ of 0.24-0.66 ($p \sim 0$) in most seasons and larger $r$ in summers.  Interannual
variations were found in simulated regional and local anthropogenic contributions, averaged at
~75% (67%-83%) and 25% (17%-33%), respectively, to wintertime NYC anthropogenically
induced GEM concentrations.  Results from this study suggest the possibility that the
increasingly strong Bermuda High over the past decades could dominate over anthropogenic
mercury emission control in affecting ambient concentrations of mercury via regional build-up
and possibly enhancing natural and legacy emissions.



## 1. Introduction

Atmospheric mercury (Hg) is a prevailing pollutant that has global consequences for both
human and ecosystem health, and hence Hg emission control is imperative.  Mercury in the
atmosphere is operationally defined in three forms, gaseous elemental mercury (GEM), gaseous
oxidized mercury (GOM), and particulate-bound mercury (PBM). Total gaseous mercury (TGM)
is the sum of GEM and GOM. The most abundant of these three forms is GEM with a lifetime of
0.5 - 1 year (Driscoll et al., 2013) and mixing ratios on the order of hundreds of parts per
quadrillion (ppqv) (~ a few ng m$^{-3}$ at 1 ng m$^{-3}$ = 112 ppqv in a standard atmosphere; unit
conversion was done in a standard atmosphere hereafter) compared to GOM and PBM with
lifetimes of hours to weeks and mixing ratios often on the order of single ppqv (~ a few pg m$^{-3}$).
The median concentration of TGM/GEM in global continental remote areas was 1.6 ng
m$^{-3}$ (180 ppqv) estimated from a large body of measurement studies (Mao et al., 2016), and the
background concentration of GEM in the Northern Hemisphere was 1.5 – 1.7 ng m$^{-3}$ (168 – 190
ppqv) (Lindberg et al., 2007).  Urban concentrations of GEM/TGM in the U.S. varied over 0.05
– 324 ng m$^{-3}$ (5.6 – 36288 ppqv) (Mao et al., 2016).   In comparison, urban concentrations and
their temporal variability were larger than rural, remote, and high elevation concentrations in the
Northern Hemisphere (e.g., Kim and Kim, 2001; Feng et al., 2003; Denis et al., 2006; Liu et al.,
2007; Peterson et al., 2009; Sprovieri et al., 2010; Zhu et al., 2012; Lan et al., 2012, 2014; Chen
et al., 2013; Civerolo et al., 2014; Fu et al., 2015; Brown et al., 2015; Mao et al., 2016 and
references therein) owing to numerous controlling factors including anthropogenic and legacy
emissions, deposition, meteorology, transport, and atmospheric chemistry (Mao et al., 2016).
Over the United States measurements from the Atmospheric Mercury Network (AMNet)
sites, located in urban, suburban, rural, and remote areas, suggested that monthly median GEM





mixing ratios varied from 148 to 226 ppqv (~1.32 to 2.02 ng m$^{-3}$) with urban values at the higher
end of the range (Lan et al., 2012).  Urban ambient atmospheric TGM/GEM concentrations in
Canada on average ranged over 1.7 – 4.5 ng m$^{-3}$ (190 – 504 ng m$^{-3}$) (Mao et al., 2016; reference
therein).  Urban GEM/TGM concentrations in Asia could be an order of magnitude larger than
those in the U.S., Canada, and Europe (Mao et al., 2016; references therein).  Many studies
identified local sources as a predominant factor controlling urban ambient concentrations
(Gabriel et al., 2005; Lyman and Gustin, 2009; Wang et al., 2013; Feng et al., 2003; Fang et al.,
2004; Zhu et al., 2012; Hall et al., 2014; Seo et al., 2016; Kim et al., 2016).  In some urban
locations nighttime daily maximums and spring-summer annual peaks were attributed to local
and regional sources followed by boundary layer dynamics and meteorological conditions (Liu et
al., 2007; Cheng et al., 2009; Liu et al., 2010; Nair et al., 2012; Zhu et al., 2012).  Surface
emissions were also suggested to play a major role in warm season annual maximums (Denis et
al., 2006; Zhu et al., 2012).  Some sites experienced early morning daily maximums with the
strongest diurnal variation in summer, due possibly to local anthropogenic sources and surface
emissions (Stamenkovic et al., 2007; Peterson et al., 2009).  Wintertime annual maximums were
attributed to less oxidation of GEM (Stamenkovic et al., 2007) and periods of cold and stagnant
air probably leading to build-up of pollution and more Hg evasion prompted by wet conditions
(Peterson et al., 2009).  Temporal variations of GEM concentrations could be attributed to the
combined influence of environmental variables, anthropogenic sources, photochemistry, and
regional transport (Xu et al., 2014).
Some studies suggested that regional sources dominated over local ones in contributing to
urban ambient Hg concentrations (e.g., Liu et al., 2007; Kim et al., 2013; Engle et al.; 2010; Xu
et al., 2014; Hall et al., 2014).  On interannual time scales, the impact of regional transport, in



comparison to local sources, could vary greatly due to large variability in atmospheric circulation
and subsequently affect urban ambient concentrations very differently.  Additional emission
control is anticipated in the future associated with the Mercury and Air Toxics Standards
(MATS) rule and the United Nations Environment Program (UNEP) international Minamata
Treaty (Selin, 2014).  To regulate future emissions, it is important to understand and quantify
contributions of local versus regional sources to urban ambient concentrations.  The objective of
this paper is to examine the seasonal, annual, and interannual variability of GEM in the Bronx
Borough of New York City (NYC) and its relation with large-scale circulation, and the
contributions of local and regional sources to NYC ambient GEM concentrations.
**2. Site Description, Measurement Data, and Analysis Methods**

The site discussed herein is maintained by the New York State Department of

Environmental Conservation (NYSDEC) as a part of AMNet under the National Atmospheric
Deposition Program (NADP).  The site is located in a densely populated area of NYC, in the
borough of Bronx at the Pfizer Plant Research Laboratory within the New York Botanical
Garden (40.8679°N, 73.8781°W).  The Bronx site is in close proximity to densely populated
local anthropogenic sources as well as downwind of many regional sources (Fig. 1).
GEM was measured every 5 minutes using a Tekran (Toronto, ON) model 2537B cold
vapor atomic fluorescence (CVAF) analyzer with a nominal detection limit of <0.1 ng m$^{-3}$ (~11.2
ppqv).  The instrument was calibrated daily with an internal permeation source.  The Tekran
system was operated according to standard operating procedures from the NADP's AMNet. The
AMNet site liaison performs annual site visits, which include manual injections to verify the
internal permeation source, and is responsible for quality assurance of the data (Civerolo et al.,
2014).  Additional details can be found in Landis et al. (2002) and Gay et al. (2013).





Measurement data of sulfur dioxide ($SO_2$), nitrogen dioxide ($NO_2$), carbon monoxide
(CO), temperature, wind direction, and wind speed were averaged hourly.  The $SO_2$
measurements were taken using a TEI 43C and a 43i TLE instrument using equivalent method
060 and pulsed fluorescence. The $NO_2$ measurements were taken using a TEI 42C instrument
using reference method 074 and chemiluminescence and catalytic conversion.  CO was measured
by a TEI 48C and an API 300EU instrument using reference method 054 with non-dispersive
infrared absorption.  The technical details of the deployment of these instruments are given by
the NYSDEC at www.dec.ny.gov/chemical/8541.html and in the 2016 Annual Monitoring
Network Plan (www.dec.ny.gov/docs/air_pdf/2016plan.pdf).
The impact of regional and local anthropogenic sources was simulated using the NOAA
Hybrid Single Particle Lagrangian Integrated Trajectory model (HYSPLIT) dispersion version
(Draxler and Hess, 1997, 1998; Draxler, 1999; Stein et al., 2015) for the winters and summers of
2009 – 2015.  HYSPLIT was driven by the EDAS 40 km model output over a domain extending
westward to OH, southward to northern VA, and northward to include New England (Fig. 10a),
with a total of 522 counties reporting Hg emissions that were extracted from the Environmental
Protection Agency's National Emission Inventory 2011 (https://www.epa.gov/air-emissions-
inventories/2011-national-emissions-inventory-nei-data ).  Two emission scenarios of 120-hour
forward dispersion simulations were conducted for each day of a season.  One included
emissions from all the 522 countries, and another excluded emissions from the five boroughs in
NYC.  The difference in NYC GEM concentrations between the two scenarios was used to
approximate the effect of anthropogenic emissions in NYC (denoted as local sources) versus
outside of NYC (denoted as regional sources) on NYC ambient Hg concentrations, not
considering loss in transit.





In the analysis of the North American trough, the trough axis index (TAI) and trough
intensity index (TII) defined by Bradbury et al. (2002) were used to quantify the position and
intensity of the North American trough.  The seasonal TAI quantifies the mean longitudinal
position of the quasi-stationary midtropospheric East Coast trough. The TAI domain extends
from 120°W to 30°W and the southern to northern boundaries ranged from 40°N to 50°N.  The
TAI index was calculated by averaging the longitudinal positions (Lon) of the minimum 500-hPa
heights ($H_{min}$) observed at each of the four latitudinal steps ($j$=40°, 42.5°, 45°, and 47.5°N)
within the index range, to produce a practical index in longitudinal units (relative to the prime
meridian):

$TAI = average\ [Lon(H_{min})_j]$

The TII is an estimate of wave amplitude at 42.5°N and is the mean height change at the 500-hPa
surface from equal distances east and west of the East Coast trough axis. It was calculated:

$TII = \{[(H_{min})_i - H_{i+30°}] + [(H_{min})_i - H_{i-30°}]\}/2$

The more negative TII is, the stronger the influence of the North American Trough would be.
The 2.5° x 2.5° reanalysis data from the National Center of Environmental Protection/National
Center of Atmospheric Research were used to calculated seasonal average TAI and TII.
Additional details about TAI and TII can be found in Bradbury et al. (2002).
**3. Results and Discussion**
**3.1 General Characteristics of Diurnal, Seasonal, and Interannual Variation**
Five-minute average GEM mixing ratios were more often larger during the warm seasons
(summer-spring) than the cool seasons (fall-winter) (Fig. 2), in agreement with previous urban
site studies (Denis et al., 2006; Liu et al., 2007; Zhu et al., 2012; Zhang et al., 2013; Civerolo et
al., 2014).  The annual cycles were not as distinct and reproducible as those in rural





environments such as southern New England, where annual maximums were found in winter and
minimums in fall (Mao et al., 2008; Sigler et al., 2009; Mao and Talbot, 2012).

Three salient features were evident in the interannual variation of a range of percentile

mixing ratios of GEM (Fig. 3; Table 1). First, the 2009 – 2010 cool season percentile values of
GEM were the lowest of all cool seasons. Second, the 2011 warm season percentile values were
the lowest of all warm seasons, even lower than in the cool season of the same year, not
reproducing the 2009 and 2010 annual cycles.  Third, the 2014 and 2015 seasonal percentile
values were mostly the highest of the study period and for the first time since 2011, warm season
values exceeded the cool season ones reproducing the 2009 and 2010 annual cycles.

The most pronounced diurnal cycles occurred in summer, as shown in seasonal average

diurnal cycles in Figure 4, with a peak between 02:00 and 06:00 UTC and a minimum between
10:00 and 16:00 UTC, which is consistent with previous studies for urban locations (e.g. Denis et
al., 2006; Liu et al., 2007; Zhu et al., 2012; Lan et al., 2012).  In summers of 2009 – 2012, the
daily maximum was ~170-190 ppqv and the daily minimum ~140-160 ppqv.  In summers of
2014 and 2015 mixing ratios were elevated greatly throughout the day with daily peaks reaching
~200 and 290 ppqv and minimums ~180 and 225 ppqv, respectively.  The diurnal amplitude,
defined as the difference between the daily maximum and minimum, was up to ~50 ppqv in
summer, ~20 ppqv in fall and spring, and <10 ppqv in winter.

During 2008 – 2013, the cool seasons experienced much larger interannual variability of

GEM than the warm seasons did, whereas in 2014 and 2015 GEM concentrations were elevated
significantly above other years in all seasons (Fig. 4).  Over 2008 - 2013, the largest interannual
variability up to ~40 ppqv difference was observed between the lowest GEM mixing ratios in fall
2009 and largest in falls 2011 and 2012, whereas springs and summers experienced much less





interannual variability except spring and summer 2011, as aforementioned, that saw the lowest
GEM mixing ratios, ~ 20 ppqv lower than all other warm seasons.
**3.2. Interannual Variation of Cool Season GEM**
The 2009 – 2010 cool season exhibited the lowest percentile values whereas most of
winter 2014 and cool season 2014-2015 percentile values were the highest of the study period
(Table 1; Fig. 3). The difference in percentile values between the two cool seasons ranged from
33-34 ppqv in the 25th and median values to 112 ppqv in the 90th percentile value.  The possible
effect of anthropogenic emission changes on those interannual variations in GEM concentrations
was the very first to be examined.  EPA national emission inventories showed a 13% decrease
from 2008 (31810 kg) to 2011 (27695 kg), and then an increase of 2% to 2014 (28270 kg) in
total emissions from the Eastern US, including states east of the Mississippi River, of which
NYC emissions increased from 125 kg in 2008 to 145 kg in 2011 and to 199 kg in 2014
(https://www.epa.gov/air-emissions-inventories/2014-national-emissions-inventory-nei-data).
Using an average PBL height of 1000 m over the Eastern US (the surface area for the Eastern US
is $2.483 \times 10^{12}$ m$^2$), a decrease of 4115 kg from 2008 to 2011 emissions was converted to ~200
ppqv and averaged at ~0.2 ppqv d$^{-1}$ over all days of the three years, and an increase of 575 kg
from 2011 to 2014 emissions was converted to ~0.03 ppqv d$^{-1}$.  The potential change in NYC
atmospheric concentrations were estimated to be ~3 ppqv d$^{-1}$ from the 2008 – 2011 NYC
emission increases alone and ~6 ppqv d$^{-1}$ from the 2011 – 2014 increase.  The potential changes
in ambient concentrations caused by the regional emission decrease/increase were negligible
compared to the observed interannual difference.  Those caused possibly by NYC emission
increases could be significant but appeared to be inconsistent with the changes in ambient
concentrations in two ways.  First, from 2010 to 2011 NYC emissions increased and yet





summertime ambient concentrations decreased by 10 ppqv throughout the seasonal averaged
diurnal cycle (Fig. 4). Second, the observed increase from summer 2012 to summer 2014 was
nearly 90 ppqv throughout the seasonal averaged diurnal cycle, a factor of 15 larger than the
effect of the local emission rise as estimated above.

Legacy and natural emissions could be another driver for the observed interannual

variations in GEM. However, seasonal mean temperature and GEM from the Bronx location
were not found to be correlated, which suggested that the effect of changes in legacy and natural
emissions on ambient GEM might not be dominant. In addition, using Zhang et al. (2016)'s
estimated annual natural and reemissions 9.4 to 13.0 μg m$^{-2}$ for the Bronx site during 2009-2014,
the maximum year-to-year change was calculated to be ~1 ppqv d$^{-1}$ assuming an average
planetary boundary layer (PBL) height of 1000 m. This change alone could not explain the
observed interannual variations. It alludes to the potential effect of *regional* legacy and natural
emissions as well as chemistry, which needs to employ modeling tools and is beyond the scope
of this study. Here, it was hypothesized that atmospheric circulation was the predominant factor
causing interannual variation in Bronx ambient GEM concentrations.

To validate this hypothesis, circulation patterns were examined first using the Bronx site

wind data. In falls 2008, 2009, 2011, and 2013, wind came from all four quadrants with
comparable frequency ranging over 15% - 30% of the season, whereas in fall 2010 the
northwesterly (270°-360°) was more frequent (37%), and the northeasterly (0°-90°) became
predominant (~50 – 74%) in falls 2012 and 2014 (Fig. 5a). The winters experienced
northwesterly winds (270°-360°) more often ranging from ~40% to 65% of the season with the
exception of winter 2015 when a little below 40% of the season on par with southwesterly wind
(180°-270°) (Fig. 5a). Wind speed was averaged seasonally for the four wind quadrants (Fig.





5b).  The strongest wind, reaching over 3 m s$^{-1}$, appeared in the northwesterly (270°-360°) in
winter and spring from 2009 to 2013.  Over 2014 and 2015 the most distinct change was
northwesterly wind speed being significantly reduced to a little over 2 m s$^{-1}$. The next in line was
southwesterly (180°-270°) wind hovering around 2 m s$^{-1}$ in the cool seasons except 2009-2010
when it was lowered to 1 m s$^{-1}$, nearly halved compared to other years.  Wind speed in the two
easterly quadrants (0°-180°) was comparable varying over 1 – 2 m s$^{-1}$ except that in spring 2013
it reached 2.5 m s$^{-1}$ and was particularly low (0.5 m s$^{-1}$) in the 2014 cool season.

Since Hg sources are mostly concentrated to the west, southwest, south, and northeast of

the Bronx site with much fewer sources to the northwest (Fig. 1), GEM mixing ratios would vary
expectedly corresponding to air masses arriving from different directions.  This was clearly
suggested by mixing ratios of GEM averaged seasonally for the four wind quadrants (Fig. 5c).
Generally, seasonally averaged GEM mixing ratios were larger by ~20-50 ppqv in the two
southerly than those in the northerly quadrants.  One exception was summer 2014 when the
average concentration in the northeasterly quadrant reached up to 275 ppqv, and the frequency of
the northeasterly was the highest at 70% with average speed less than 1 m s$^{-1}$.  Such weak wind
indicated fairly calm conditions in the region and the wind direction data were not meaningful.
Overall, in addition to local emissions, interannual variability in the origin of the air masses
reaching Bronx appeared to cast significant influence on the ambient concentrations of GEM in
the city.

Two cases, the lowest percentile values in the 2009–2010 cool season and the highest in

2014-2015, were used to elaborate on this point.  What stood out in the 2009 – 2010 cool season
was very low frequency (14%) of wind from the southwesterly quadrant (180°-270°) in fall 2009
and the largest frequency of wind from the northwesterly quadrant (67%, 270°-360°) in winter



245 2010 combined with nearly the lowest wind speed ($\leq 1$ m s$^{-1}$) in the three quadrants (0°-270°)

246 (Figs. 5a-c).  This indicated that the particularly low mixing ratios in the cool season of 2009 –

247 2010 were likely caused by over 4 times more frequent influx of relatively cleaner Canadian air

248 masses and slowest southerly flow of more polluted air.

249  The second case is winter 2014 when GEM averaged in the four wind quadrants reached

250 the maximums of all time respectively (Fig. 5c).  Coincidently the frequency of wind from the

251 northwesterly quadrant (270°-360°) was nearly the lowest of all cool seasons barely reaching

252 40% of the season compared to up to 67% in winter 2010 (Fig. 5a).  Meanwhile, the frequency of

253 wind from the southwesterly quadrant (180°-270°) reached a high of 34% of all cool seasons,

254 and the wind speed of ~2 m s$^{-1}$ was comparable to the northwesterly.  This is a strong indication

255 of arrival of air masses rich in GEM originating from the heavy emitters in the Northeastern U.S.

256 Urban Corridor via flow nearly as frequent and as fast as the relatively clean northwesterly.

257 Winter 2015 showed similar wind patterns, also coincided with high GEM concentrations.

258  Such variations in wind direction and speed at the Bronx site can be better understood in

259 the context of large-scale circulation. The climatological 500 hPa geopotential height (GPH)

260 (1980-2010) for cool seasons during 1980-2010 exhibited the North American trough centered

261 over coastal southeastern Canada extending southwestward over the Eastern US (Fig. 6a).  All

262 cool seasons experienced variations of this pattern except cool seasons 2009-2010 and 2013-

263 2014 that appeared to be anomalous (Figs. 6b and 6c).  Specifically the trough in winter 2010

264 shifted eastward farthest out over the ocean and was the weakest, evidenced in the maximum

265 TAI (62°W) and nearly the least negative TII value (-80 m) (Fig. 6d).  In contrast, the trough in

266 winter 2014 was situated the farthest over land and the strongest of all winters, backed by the

267 most negative TAI (85°W) and nearly the most negative TII value (-201 m) (Fig. 6d) .  This



suggested that in winter 2010 the Northeast US was most frequently under the influence of air
masses from higher latitudes via flow on the backside of the North American trough whereas
much less so due to the East U.S. positioned near the axis to the front of the trough in winter
2014.  This was further clearly reflected in the maps of sea level pressure (SLP) for the two
winters.  The unusual winter 2010 circulation was signified by northerly gradient flow (Fig. 6f)
from the backside of the Icelandic Low, which shifted toward the south and west near
Newfoundland compared to its 1980-2010 climatological position right between and below
Greenland and Iceland (Fig. 6e).  This indicated predominant transport of relatively clean air
from Canada combined with strong ventilation of continental pollution, likely leading to the least
polluted air in winter 2010 of all 7 winters.  In contrast, in winter 2014 NYC appeared to be on
the periphery of high pressure systems in predominantly slow northwesterly and southwesterly
flow regimes (Fig. 6g).  This explains the least frequent, lowest wind speed in the easterly wind
quadrants during winter 2014 (Fig. 5b), which is conducive to regional build-up of air pollution,
resulting in the highest mixing ratios of GEM of all winters.  More evidence was shown in
Section 6 using modeled contributions to NYC ambient concentrations from local versus
regional anthropogenic sources.
**3.3 Interannaul Variation of Warm Season GEM**
**3.3.1 Annual maximums in warm seasons of 2009 and 2010**

The annual cycles of GEM at the Bronx site in 2009 and 2010, with larger values in

spring and summer (Table 1; Figs. 2 and 3), is consistent with measurements from some urban
and industrial locations in the literature (Lindberg and Stratton, 1998; Liu et al., 2007; Zhu et al.,
2012; Xu et al., 2014).  Lindberg and Stratton (1998) and Liu et al. (2007) attributed such annual
cycles to local anthropogenic sources, while Zhu et al. (2012) and Xu et al. (2014) speculated



reemission from soils to be a potential dominant factor. In NYC, impervious surfaces comprise
95% of the total land surface (Adler and Tanner, 2013), which, considering local sources alone,
makes reemission of Hg from soils much less significant than anthropogenic emissions from the
area. Indeed no correlation between seasonal temperature and GEM was found for the Bronx
site as mentioned in Section 3.2. It thus seemed unlikely that NYC legacy emissions contributed
to the 2009 and 2010 annual cycles. The impact of regional vs. local anthropogenic sources on
NYC GEM concentrations was studied in Section 6, and quantifying the impact of *regional*
natural and legacy emissions calls for a regional modeling approach, which is beyond the scope
of this study. Here we focused on the potential impact of circulation on NYC GEM
concentrations.

In the warm seasons, Bronx was on the periphery of the Bermuda High in transition to

under the influence of the North American trough, and consequently Bronx was, as most of the
eastern US was, more frequently under the high pressure system influence (Figs. 7c,f), which is
usually lower wind speed. This is consistent with the annual cycle of wind speed shown in
Figure 5b, with wind mostly lower in spring-summer and higher in fall-winter conducive to
regional pollution build-up, which could explain why Bronx saw larger peaks of GEM in the
warm season than in the cool season.

**3.3.2 Lowest GEM in warm season 2011 and highest in 2014**

In examining wind in the warm season 2011, what stood out was that Bronx experienced

significantly increased frequency (37%) of northeasterly wind at wind speed nearly 2 m s$^{-1}$ in
spring and decreased frequency of (20%) of northwesterly wind in summer compared to the
spring and summer in 2009 and 2010 (Fig. 5a). In summer 2014 nearly 80% of the season had
northeasterly wind (0°-90°) and there was unusually weak wind (~1 m s$^{-1}$) in all four wind



quadrants (Figs. 5a,b), which suggested calm conditions.  In summer 2011GEM concentrations
in the northeasterly wind quadrant were averaged ~145 ppqv, ~ 30 ppqv lower than that in the
most polluted southerly quadrants (Fig. 5c).  In contrast, summer 2014 GEM in the northeasterly
quadrant was averaged 275 ppqv compared to ~160 – 200 ppqv in the other three quadrants (Fig.
5c).  The unusually high concentration was an indication of build-up under calm conditions.

The anomalously increased occurrence of northeasterly wind in summer 2011 indicated

unusual circulation.  Compared to the 1980-2010 climatology, the 500 hPa GPH in spring 2011
showed the weakest North American trough of all springs (Fig. S1), evidenced in the
westernmost trough axis position (TAI = 108°W) and the smallest intensity (TII = -27 m) of all
springs (Fig. 7a).  The 500 hPa GPH in summer 2011 suggested the strongest North American
trough (TII = -87 m) and the second easternmost trough axis position (TAI = 66°W) of all
summers (Fig. 7b; Fig. S2).  This suggested that the Northeast U.S. in summer 2011 was
frequently under significant influence of the backside of the trough, i.e. sweeping air flow from
higher latitudes subsiding to the surface in midlatitudes.

Near the surface, the 1980-2010 SLP climatology suggested that in spring NYC was

situated in the gradient flow of the Bermuda High and a trough from the Icelandic Low (Fig. 7c),
conducive to transport of emissions from upstream source regions such as upstate New York,
Ohio (OH) and Pennsylvania (PA), while in summer under the influence of the Bermuda High
favorable to regional build-up (Fig. 7f).  However, in spring 2011, the trough of the Icelandic
Low gave way to the Canadian High leaving NYC locked in a zone between the Canadian High
and subtropical high (Fig. 7d), possibly cutting regional transport short in addition to strong
subsidence of cleaner higher latitudinal air leading to the lowest concentrations of GEM of all
springs.  Similarly unusual was summer 2011 when NYC was under less influence of the



Bermuda High than that of the North American Trough unfavorable to regional build-up (Figs.
7g).  These speculations appeared to be consistent with the fact that both seasons saw unusual
equal chances of winds from the four quadrants (Fig. 5a) over Bronx and its surrounding areas.

The 500 hPa TAI and TII values (Figs. 7b) and the 500 hPa GPH map in summer 2014

(Fig. S2) suggested the weakest North American trough (TII = -44 m) of all summers, with its
axis on average at 72°W, near the East Coast.  This indicates that summer 2014 experienced the
strongest influence of the Bermuda High on the East Coast (Figs. 7e,h) of all summers during the
study period, the polar extreme of the 2011 warm season.  Corresponding to that, the summer
2014 SLP map (Fig. S4) exhibited the Bermuda High ridge over the Eastern U.S. more north-
extending than in other years, which is consistent with weak winds in all directions as shown in
Fig. 5b.  This dynamic situation led to regional build-up conducive to the highest GEM mixing
ratios in all wind quadrants.

To be quantitative, domain (25°N-50°N, 95°W-70°W) average SLP was used as an

indicator to gauge the influence of the Bermuda High, and the number of grids with SLP
exceeding 1014 hPa over the domain, the northernmost latitude, and westernmost longitude of
the 1014 hPa isobar were used to gauge the horizontal spatial extent of the influence of the
Bermuda High.  These four indices for the summers of 2009 – 2015 were plotted together with
summertime median mixing ratios of GEM in Figure 8.  The summertime median mixing ratio of
GEM was correlated with the four indices at $r$ ranging over $0.84 - 0.96$ ($p = 0.06 - 0.009$), best
correlated with the northernmost latitude that the 1014 hPa isobar reached at $r=0.96$ ($p=0.009$).
The lowest GEM in summer 2011 was associated with the weakest influence of the Bermuda
High indicated  by its smallest spatial extent, reflected in the lowest domain averaged SLP (1013
hpa), the fewest grids with SLP > 1014 hPa (44), and the southernmost latitude (37°N) the 1014



hPa isobar reached.  The trough over the East Coast reached its southernmost point down to
North Carolina compared to other summers (Fig. S4), indicating more widespread influence of
relatively clean Canadian air on the Eastern US sweeping out the heavily polluted air in the
region.  One may argue that the positive correlations shown above appeared to be driven by one
point in summer 2014, due to missing/unavailable data in summers 2013 and 2015.  It should be
noted that the dramatic increase in GEM in summer 2014 needs to be put in the perspective of
the seasons proceeding and following summer 2014, when GEM was increased consistently in
seasons from winter 2014 through spring 2015 compared to all previous years.  Therefore, the
large increase in summer 2014 was most likely not fortuitous, and more importantly such
increases were consistent with the driving physical mechanisms as suggested in the large scale-
circulation.

It should be noted that the seasonal median GEM values in the four wind quadrants

exhibited trends largely consistent with those in the overall seasonal values (r=0.71 – 0.93, $p$ ~0),
and the ones in the more polluted southerly quadrants were slightly more so (r=0.93, $p$ ~0) (Fig.
5c).  This suggests that changes in ambient mixing ratios occurred in air masses coming from all
directions, whether they were from the relatively clean northwest and northeast, or the heavily
polluted regions southeast and southwest of Bronx.  This was perhaps because the lifetime of
GEM is long enough for air from all wind directions to be regionally mixed.  Overall, the fact
that the GEM values in the two relatively more polluted quadrants were correlated with the
overall values suggested that the trend in the ambient GEM mixing ratio was possibly associated
in large part with changes in anthropogenic emissions.
**4. Relationships Between GEM and Anthropogenic Tracers**

Correlations between Hg and several tracers (e.g., CO, $SO_2$, and $NO_2$) have been



commonly used to identify Hg anthropogenic sources, source-receptor relationships, and/or
emission ratios.  The linear correlation between CO and GEM, especially in winter, in rural
locations despite their different sources, reflects their emission ratios in regionally well-mixed air
masses (e.g., Mao et al., 2008).  At the Bronx site, seasonal GEM and CO were found to be
correlated with $r$ up to 0.66 ($p{\sim}0$) in all seasons over 2008 -2013, indicating significant, year-
round *regional* influence, and the two were notably not correlated in all the seasons from winter
2014 through spring 2015.  Over 2008 – 2013 $r^2$ values of GEM-CO were larger in warm than in
cold seasons with the maximums exceeding 0.40 in spring 2010 and winter – summer 2011(Fig.
9). The slope value varied from the smallest (~0.14 ppqv ppbv$^{-1}$) in spring-summer 2010 to the
largest (0.21 ppqv ppbv$^{-1}$) in summer 2012 (Fig. 9), close to and higher than the upper end of the
range, 0.06 – 0.14 ppqv ppbv$^{-1}$, from rural southern New Hampshire (NH) during winters 2004 –
2007 (Mao et al., 2008).  This was greatly different from the GEM-CO correlation in rural
southern NH in winter only due to confounding factors such as legacy emissions and wet
deposition in summer (Mao et al., 2008; Lombard et al., 2011).  Bronx experiencing more
significant GEM-CO correlation in warm seasons indicated better regionally mixed air masses,
influenced predominantly by anthropogenic emissions, than in cool seasons.  This is consistent
with the cool and warm seasonal circulation patterns as discussed in Sections 3.2 and 3.3, which
is that in warm seasons NYC was predominantly under the influence of the subtropical high
conducive to regional mixing and build-up of pollutants.

No correlation between GEM and CO over 2014 – 2015 could be due in part to the more

dramatic emission reductions in CO than changes in GEM in the Eastern U.S.  The high
percentile values of CO at the Bronx site had been affected by anthropogenic emission
reductions over the years, while the 10[th] and 25[th] percentile values remained fairly constant in all





seasons (Fig. S5).  Zhou et al. (2015) suggested that baseline CO in Northeastern US rural areas
was controlled by a multitude of factors including global biomass emissions, large-scale
circulation, and cyclone activity.  At the Bronx site, the low percentile value, close to regional
baseline levels, was possibly determined by a range of factors, whose importance could have
varied from year to year.

Unlike previous studies (e.g., Jen et al., 2013; Choi et al., 2013), GEM at the Bronx site

was found hardly correlated with $SO_2$ while somewhat to moderately correlated with $NO_2$ ($r =$
$0.22 - 0.64$, $p<0.0001$) (Table 2), despite abundant sources co-emitting GEM, $SO_2$, and $NO_2$
locally and upwind.  In addition to different lifetimes, different magnitude and timing of
emission reduction implementations and source types of the three compounds could have
affected their relation.  Total Hg anthropogenic emissions in NYC were increased by 16% from
2008 to 2011, mainly in miscellaneous non-industrial NEC and waste disposal emissions, and
further increased by 37% from 2011 to 2014 primarily in fuel combustion. As aforementioned,
emissions of Hg in the Eastern U.S. decreased by 13% from 2008 to 2011 and increased by 2%
from 2011 to 2014.   In contrast, total $SO_2$ emissions in NYC decreased steadily by 30% from
2008 to 2011 followed by a further decrease of 43% to 2014, while over the Eastern U.S
decreased by 48% from 2008 to 2011 and furthered by another 29% decrease in 2014.  The effect
of these decreases in $SO_2$ emissions was reflected in the Bronx data, with a 58% decrease in the
seasonal median mixing ratio of $SO_2$ from 9.2 ppbv in winter 2009 to 2.8 ppbv in winter 2015
(Fig. S6).  As for $NO_2$, fuel and mobile combustion emissions comprised >99.5% of the total
$NO_x$ emissions in NYC and ~90% over the Eastern US.  NYC $NO_x$ emissions changed
insignificantly (1%) from 2008 to 2011 and by 15% from 2011 to 2014, while Eastern US
mobile and fuel combustion emissions were decreased by 16% and 33%, respectively, from 2008





to 2011, and further decreased by 13% and 9%, respectively, to 2014.  These varying changes
possibly contributed to confounding the emission signature of GEM vs. $NO_x$ and altered that of
GEM vs $SO_2$.

The effect of local emissions can be accentuated by the correlation between GEM and

$SO_2$ and between GEM and $NO_2$ for the $SO_2$ and $NO_2$ mixing ratios exceeding their respective
seasonal 95[th] percentile concentrations.  However, nearly no correlation between GEM and $SO_2$
as well as between GEM and $NO_2$ was found in this subset of data (Table 2).  It should therefore
be cautioned that tracer correlation could not be used to identify source types of GEM or
estimate emission ratios of GEM to $SO_2$ or $NO_2$ in NYC.
**5. Regional vs. Local Contributions to NYC Ambient GEM Concentrations**

HYSPLIT dispersion simulations were used to obtain a quantitative comparison of the

effects of sources outside and inside NYC on NYC ambient concentrations of GEM.   As stated
in Section 2, the modeling domain extended westward to OH and southward to northern VA, and
northward to include New England (Fig. 10a), with a total of 522 counties reporting Hg
emissions.  Shown in Figure 10b is the contribution, in percentage of the total contribution from
all anthropogenic emissions in the domain, to NYC ambient concentrations of GEM from
anthropogenic emissions alone from local sources, and in Figure 10c is the contribution of
emissions from regional anthropogenic sources.  There was clearly interannual variability in the
contribution of local versus regional anthropogenic sources.  Local emissions averaged a
contribution of 25% in all winters of 2009 – 2015 with the period minimum of 17% in winter
2011 and the maximum of 33% in winter 2013 (Fig. 10b).   Conversely, the contribution of
regional sources averaged a contribution of 75% in all winters with the largest 83% in winter
2011 and the lowest 67% in winter 2013 (Fig. 10c).  Compared to that in the winter of the same



year, contributions from local sources were larger (by up to 12% in 2009) in summer 2009, 2011,
2012, and 2014, close in summer 2010, and 10% smaller in summer 2013 (Fig. 10b).

A close examination revealed largely consistent relation between NYC GEM mixing

ratios and source contributions. As suggested in Section 3.2, Bronx in winter 2010 experienced
the lowest concentrations of GEM in all percentile values, and yet, interestingly the simulated
local contribution in winter 2010 was in the mid-range of the 7 winters. This indicates that the
particularly low background concentration in the sweeping northerly flow led to less regional
contribution to NYC Hg concentrations than regional sources would in other years. In contrast,
winter 2014 saw the highest $25^{th}$, $50^{th}$, $75^{th}$, and $90^{th}$ percentile concentrations of GEM, and yet
the contribution of local sources (~22%) was not even higher than average (25%). As
aforementioned, in winter 2014 the Eastern U.S. was most likely under the least dynamic
conditions conducive to regional build-up of air pollution, which resulted in a higher than
average contribution from regional sources and conversely lower than average contribution from
local sources (Fig. 10c). Consistent with GEM, the lower percentile mixing ratios of CO, $SO_2$,
and $NO_2$ appeared to be elevated or stopped decreasing compared to those in the previous year
(Figs. S5,S6).

The HYSPLIT dispersion model simulations suggested that close to three quarters of the

anthropogenically induced concentration of GEM in NYC was from regional sources. It should
be noted that to save computational time, the simulation domain used in this study was smaller
than ideal. With a larger regional domain, the significance of regional anthropogenic sources
could be enhanced. In addition, other factors/processes might have competed with the effect of
anthropogenic emission reductions, such as legacy and natural emissions, deposition,
meteorology, and/or large-scale circulation. Nearly 90% of the model simulation domain is





covered by vegetation.  SMOKE model output in Ye et al. (2017) suggested that the ratio of
anthropogenic to legacy and natural emissions was 0.3 over the domain.  Legacy and natural
emissions could become dominant under warmer and wetter conditions in summer.  Moreover,
Hg deposition could be impacted by changes in physical parameters such as light, temperature,
and plant species (Rutter et al., 2011).  Indeed changes of -30% to 50% in Hg deposition were
simulated for the Eastern US from the 2000s to the 2050s due to changes in precipitation
(Megaritis et al., 2014).  Net GEM surface emissions were estimated to be dominant in summer
and net dry deposition in other seasons at majority of AMNet monitoring sites in eastern North
America (Zhang et al., 2016).  Since Hg deposition and legacy emissions are closely linked,
these studies indicate potential changes in legacy emissions in response to variable
meteorological conditions and changing climate with subsequent effects on atmospheric Hg
concentrations.   Therefore, with legacy and natural emissions accounted for, regional
contributions to NYC ambient Hg concentrations would be even more dominant.
**6. Summary**

For the Bronx site in NYC, distinct annual cycles of GEM were found in 2009 and 2010

with higher concentrations in warm than in cool seasons by 10 – 20 ppqv (~10% – 25%),
consistent with urban annual cycles reported in the literature.  This annual cycle was not
reproduced in 2011 with anomalously low concentrations in the warm season and occurred again
in 2014 with significantly enhanced annual amplitude up to ~70 ppqv.  Such temporal variability
in the urban GEM concentration was found to be driven by that in large-scale circulation.
Seasonal median mixing ratios of GEM was found to be correlated with both the North
American TAI and TII in winter and with TII in summer.  Further, the intensity and position of
the Bermuda High pressure system had a significant impact on Bronx GEM concentrations in





warm seasons.  This was evidenced in the strong correlation ($r= 0.84 – 0.96$, $p = 0.06 – 0.009$) of
seasonal median mixing ratios of GEM with four Bermuda High intensity indices, best correlated
at $r=0.96$ ($p=0.009$) with the northernmost latitude that the 1014 hPa isobar reached.  The year of
2014 experienced anomalously strong influence of the Bermuda High resulting in the largest
GEM mixing ratios of the entire study period in all percentile values throughout the year.  The
regional influence on GEM concentrations in Bronx was corroborated by significant, year-round
GEM-CO correlation ($r$ up to 0.66, $p \sim 0$) over 2008 - 2013.  This correlation disappeared
completely from winter 2014 through spring 2015 possibly resulting from their very different
emission changes in the Eastern U.S.

HYSPLIT dispersion model simulations suggested that regional sources outside of NYC

contributed to ~75% (67% - 83%) of the anthropogenic portion of the ambient GEM
concentration and NYC emissions the remaining ~25% (17% - 33%).  Significant interannual
variation in the regional and local contributions was found to be consistent with that in large-
scale circulation.  The fact that there was no clearly defined trend in GEM concentrations at the
Bronx site during the study period, despite decreases anthropogenic emission reductions in the
Eastern U.S. from 2008 to 2014, suggested that other factors/processes, such as large-scale
circulation and legacy/natural emissions, might have dominated over anthropogenic emission
reductions.

The North Atlantic Subtropical High over 1978 – 2007 had reportedly become more

intense, and its western ridge had displaced westward with an enhanced meridional movement
(Li et al., 2011). The increasing intensity and spatial extent of the high pressure system could
cast a strong influence on the Northeastern US with subsequent effect on ambient concentrations
of Hg via regional build-up and changing legacy emissions.  This could dominate over the effect





of anthropogenic emission reductions, as suggested by this study. Indeed Zhu and Liang (2013)
recommended that strong decadal variations in the Bermuda High should be considered in the
U.S. air quality dynamic management. Therefore controlling urban ambient concentrations of
Hg needs to account for the overall impact of multiple factors, which may not be dominated by
emission reductions.
**Acknowledgments**
This work was funded by the Environmental Protection Agency Grant Agreement
#83521501. We are grateful to M. L. Olson and T. R. Bergerhouse of NADP & University of
Illinois at Urbana-Champaign and K. Civerolo of NYS DEC for making the Bronx GEM data
available. We also thank K. Civerolo for helpful comments. The authors gratefully
acknowledge the NOAA Air Resources Laboratory for free access to HYSPLIT. The
measurement data used in this study could be obtained from AMNet of NADP
(http://nadp.sws.uiuc.edu/amn/data.aspx).

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










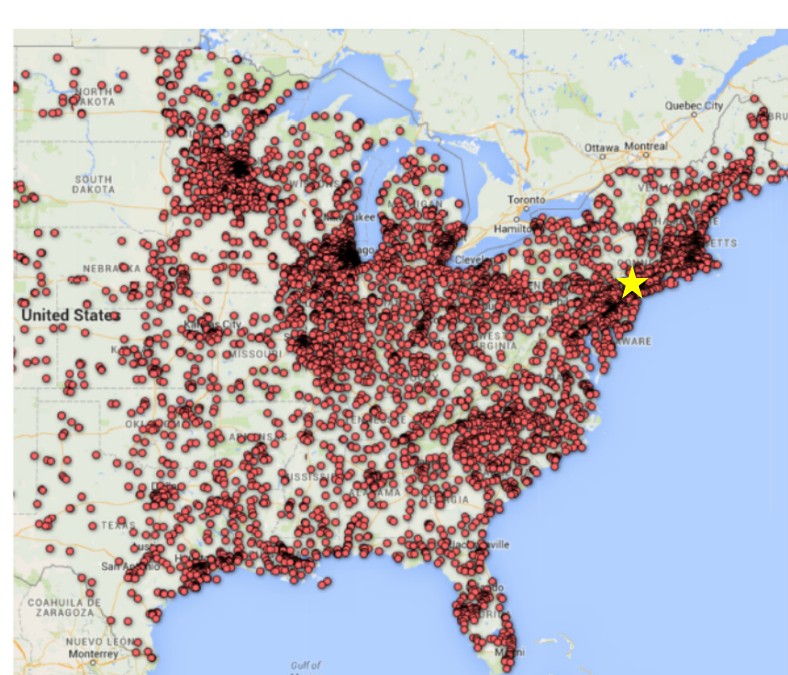

Figure 1. Map of mercury emission sources in the Eastern US.
The yellow asterisk marks the location of the Bronx site.








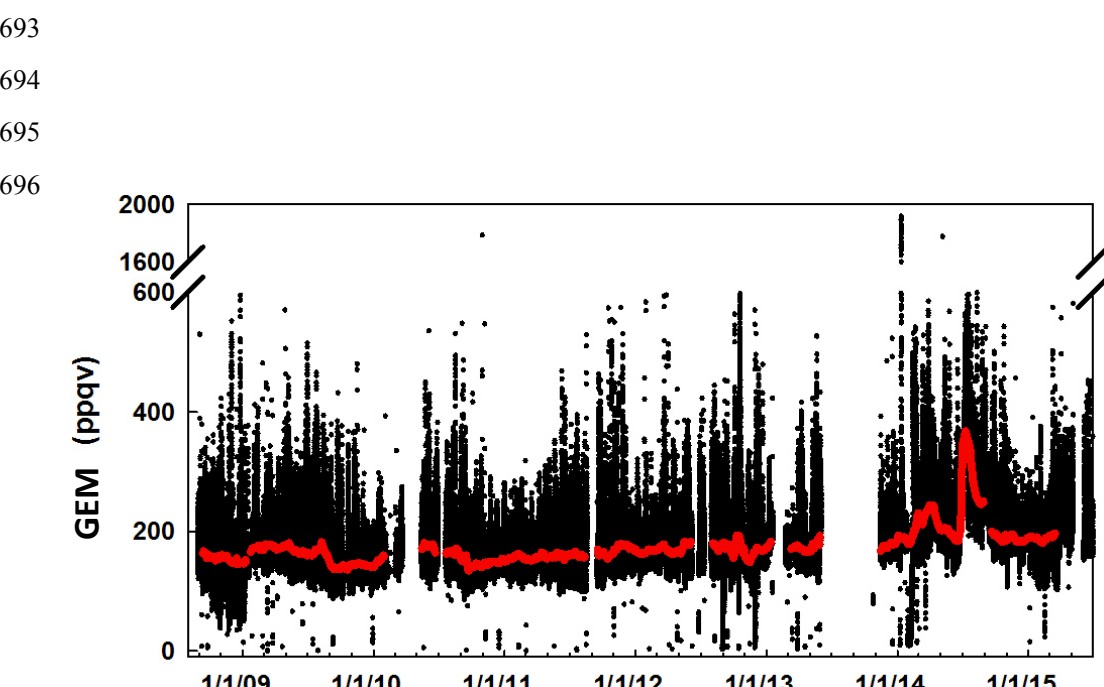

Figure 2. Time series of 5-min average GEM mixing ratios (black dots) with 30-day running average (red line) during the study period.





202















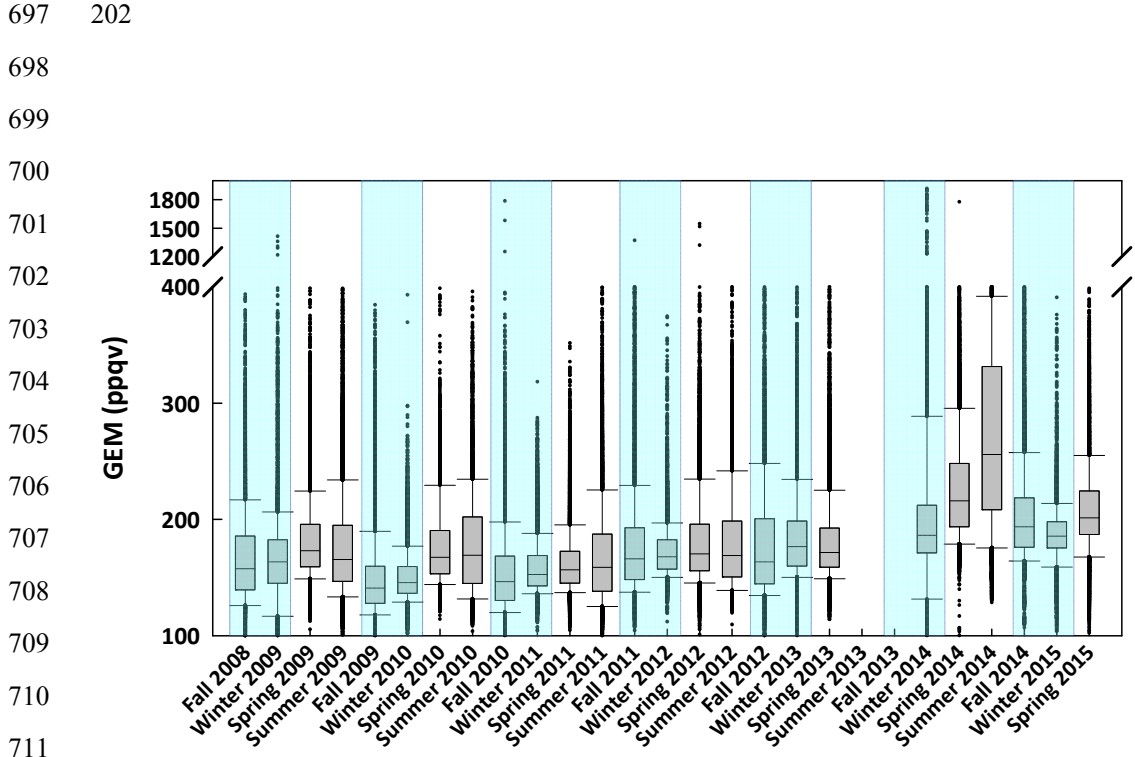

Figure 3. Seasonal 10th, 25th, 50th, 75th, and 90th GEM percentile values.  The blue shaded
areas are the cool seasons.










Figure 4. Seasonal averaged diurnal cycles of GEM for spring, summer, fall, and winter.





Figure 5. a) Fraction of wind coming from, b) wind speed, and c) TGM averaged
in the four wind quadrants in each season.  The shaded areas indicate the cool
seasons.  In b) the black dotted line indicates the wind speed averaged in all
directions. In c) the black dotted line and black solid dots represent the overall
median values of GEM.







Figure 6. September – February 500 hPa GPH for (a) 1980–2010, (b) 2010, and (c) 2014; North American trough TAI and TII for winters 2009–2015 (d); sea level pressure averaged over winters of 1980– 2010 (e), winter 2010 (f) and winter 2014 (g). The red asterisks indicate the location of the Bronx site. (Courtesy: NOAA ESRL PSD Interactive Climate Analysis).





Figure 7. The axis position (TAI) and intensity (TII) of the 500 hPa North American Trough in spring (a) and summer (b). Sea level pressure (SLP) in spring (c) 1980-2010, (d) 2011, and (e) 2014. SLP in summer (f) 1980 – 2010, (g) 2011, and (h) 2014. The red asterisks indicate the Bronx site location. (Courtesy: NOAA ESRL PSD Interactive Climate Analysis)





807

808

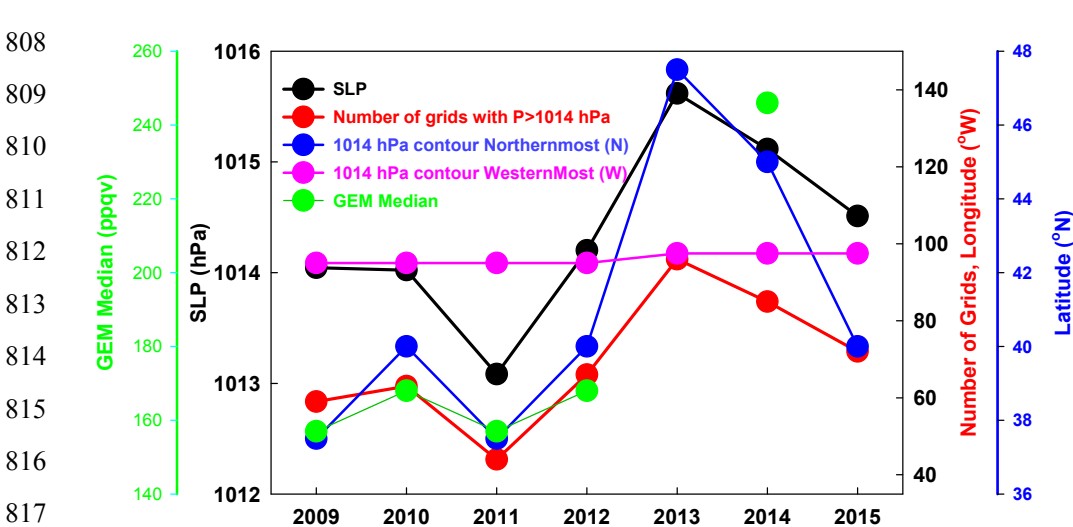

Figure 8. Average sea level pressure (SLP) over the domain of 25°N-50°N, 95°W-70°W (black), number of grids with SLP > 1014 hPa (red), the northernmost latitude (blue) and the westernmost longitude (magenta) the 1014 hPa SLP contour reached, and seasonal median GEM mixing ratios (green) in the summer.






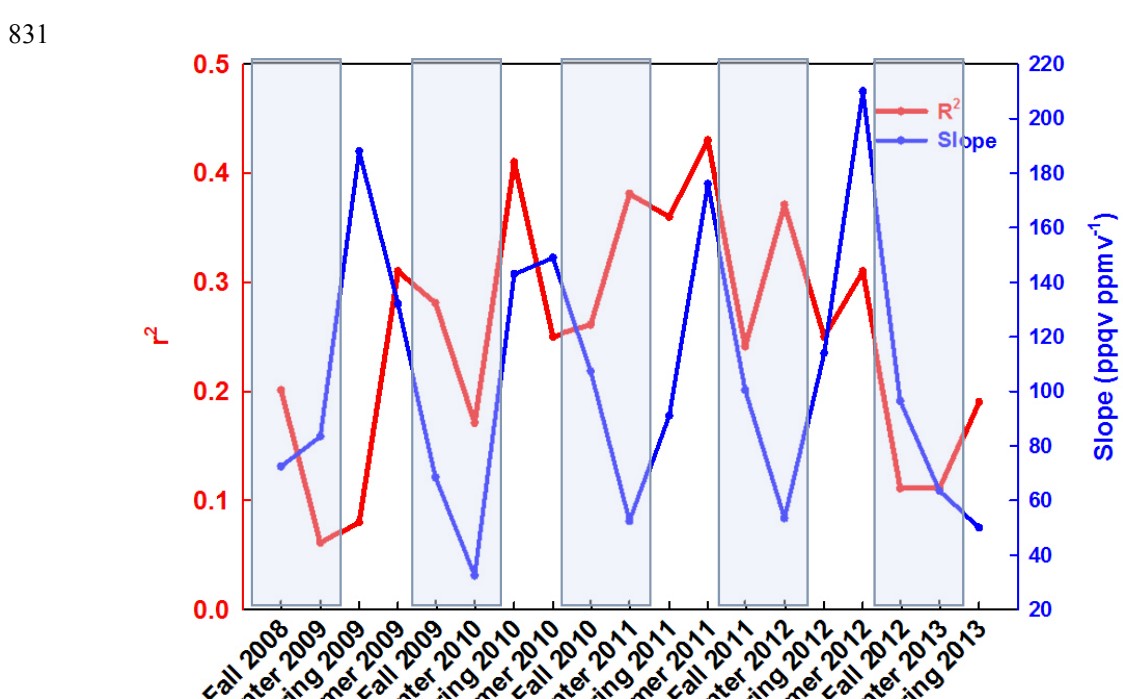

Figure 9. Values of $r^2$ (red) and slope (blue) of GEM-CO correlation during each season from 2008 to 2013. All $r^2$ values were statistically significant with $p$ approaching 0.









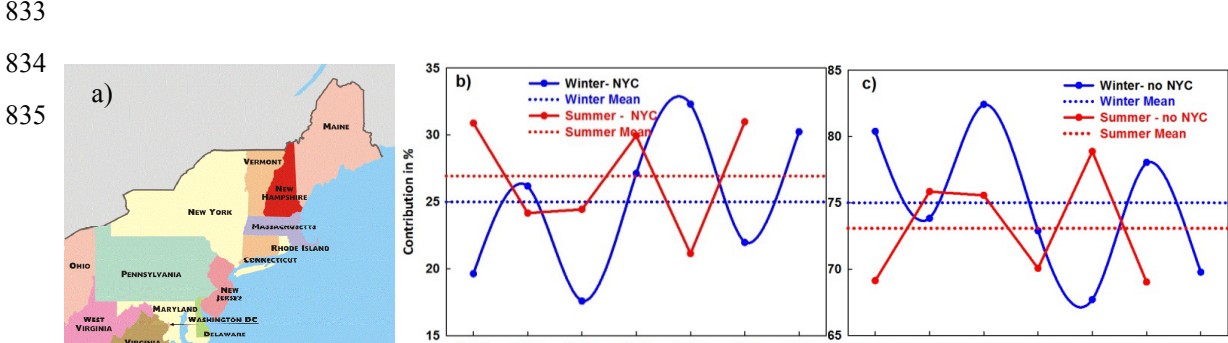

Figure 10. a) Counties and states that contributed to Hg in NYC; b) Contributions (in %) of NYC sources to NYC Hg concentrations; c) Contributions (in %) of sources outside of NYC to NYC Hg concentrations in winter (blue) and summer (red).



Table 1. Seasonal statistic metrics of GEM mixing ratios from the Bronx site.

| | | 10th | 25th | Median | 75th | 90th | Range | Sample # |
|---|---|---|---|---|---|---|---|---|
| 2008 | Fall | 123 | 134 | 157 | 179 | 213 | 22-963 | 14900 |
| 2009 | Winter | 112 | 134 | 157 | 179 | 202 | 11-7302 | 9861 |
| | Spring | 146 | 157 | 168 | 190 | 224 | 11-15310 | 15129 |
| | Summer | 123 | 146 | 157 | 190 | 224 | 11-515 | 15577 |
| | Fall | 112 | 123 | 134 | 157 | 179 | 11-470 | 13291 |
| 2010 | Winter | 123 | 134 | 146 | 157 | 168 | 11-392 | 10414 |
| | Spring | 134 | 146 | 157 | 190 | 224 | 56-448 | 4383 |
| | Summer | 123 | 134 | 168 | 201 | 224 | 78-526 | 9303 |
| | Fall | 112 | 123 | 146 | 168 | 190 | 11-1781 | 12245 |
| 2011 | Winter | 134 | 134 | 146 | 168 | 179 | 11-314 | 15364 |
| | Spring | 134 | 134 | 146 | 168 | 190 | 34-784 | 15653 |
| | Summer | 123 | 134 | 157 | 179 | 224 | 22-526 | 12548 |
| | Fall | 134 | 146 | 157 | 190 | 224 | 22-3886 | 12949 |
| 2012 | Winter | 146 | 157 | 168 | 190 | 224 | 11-1064 | 15264 |
| | Spring | 134 | 146 | 168 | 190 | 224 | 11-1546 | 13888 |
| | Summer | 134 | 146 | 168 | 190 | 235 | 11-2766 | 8406 |
| | Fall | 134 | 142 | 157 | 190 | 246 | 11-896 | 13210 |
| 2013 | Winter | 146 | 157 | 168 | 190 | 224 | 11-1064 | 7905 |
| | Spring | 146 | 157 | 168 | 190 | 224 | 11-526 | 12124 |
| | Summer | - | - | - | - | - | - | - |
| | Fall | - | - | - | - | - | - | - |
| 2014 | Winter | 134 | 168 | 179 | 202 | 280 | 11-1915 | 14709 |
| | Spring | 168 | 190 | 212 | 246 | 291 | 34-1770 | 15317 |
| | Summer | 168 | 202 | 246 | 325 | 392 | 123-1064 | 13161 |
| | Fall | 157 | 168 | 190 | 213 | 246 | 101-538 | 12847 |
| 2015 | Winter | 157 | 168 | 179 | 190 | 213 | 11-381 | 15250 |
| | Spring | 157 | 179 | 190 | 224 | 246 | 101-795 | 11124 |





Table 2. Pearson correlation coefficients (r) between GEM and $SO_2$ and between GEM and $NO_2$
with *p* values in parenthesis, for seasons during fall 2008 - spring 2015.

|  | All data | | $SO_2$ & $NO_2$ > 95th percentile | |
|---|---|---|---|---|
|  | $SO_2$ | $NO_2$ | $SO_2$ | $NO_2$ |
| Fall 2008 | -0.04 (=0.0675) | 0.28 (<0.0001) | 0.08 (=0.471) | -0.03 (=0.810) |
| Winter 2009 | 0.14 (<0.0001) | 0.29 (<0.0001) | -0.21 (=0.0877) | -0.05 (=0.687) |
| Spring 2009 | 0.08 (=0.0006) | 0.22 (<0.0001) | -0.02 (=0.832) | 0.09 (=0.41) |
| Summer 2009 | 0.15 (<0.0001) | 0.43 (<0.0001) | 0.50 (<0.0001) | 0.25 (=0.0152) |
| Fall 2009 | 0.22 (<0.0001) | 0.50 (<0.0001) | -0.29 (=0.0067) | -0.03 (=0.776) |
| Winter 2010 | 0.36 (<0.0001) | 0.49 (<0.0001) | 0.06 (=0.592) | 0.07 (=0.569) |
| Spring 2010 | 0.03 (=0.549) | 0.34 (<0.0001) | -0.13 (=0.507) | 0.13 (=0.505) |
| Summer 2010 | 0.10 (<0.0001) | 0.45 (<0.0001) | -0.07 (=0.605) | 0.22 (=0.105) |
| Fall 2010 | 0.09 (=0.0005) | 0.35 (<0.0001) | -0.01 (=0.929) | -0.06 (=0.567) |
| Winter 2011 | 0.44 (<0.0001) | 0.64 (<0.0001) | -0.23 (=0.0267) | -0.04 (=0.670) |
| Spring 2011 | 0.01 (=0.706) | 0.36 (<0.0001) | 0.01 (=0.901) | 0.24 (=0.0156) |
| Summer 2011 | 0.15 (<0.0001) | 0.48 (<0.0001) | -0.12 (=0.285) | 0.11 (=0.296) |
| Fall 2011 | 0.20 (<0.0001) | 0.48 (<0.0001) | 0.19 (=0.0839) | 0.20 (=0.0685) |
| Winter 2012 | 0.18 (<0.0001) | 0.51 (<0.0001) | 0.11 (=0.285) | 0.11 (=0.278) |
| Spring 2012 | 0.10 (<0.0001) | 0.40 (<0.0001) | -0.12 (=0.268) | -0.16 (=0.125) |
| Summer 2012 | 0.13 (<0.0001) | 0.24 (<0.0001) | -0.10 (=0.475) | 0.02 (=0.903) |
| Fall 2012 | -0.18 (<0.0001) | 0.08 (=0.0005) | -0.34 (=0.0011) | -0.37 (=0.0004) |
| Winter 2013 | 0.09 (=0.0021) | 0.37 (<0.0001) | -0.22 (=0.102) | -0.29 (=0.0282) |
| Spring 2013 | -0.02 (=0.478) | 0.43 (<0.0001) | 0.03 (=0.786) | 0.38 (=0.0004) |
| Summer 2013 | N/A | N/A | N/A | N/A |
| Fall 2013 | N/A | N/A | N/A | N/A |
| Winter 2014 | -0.15 (<0.0001) | 0.01 (=0.638) | -0.25 (=0.0154) | 0.05 (=0.638) |
| Spring 2014 | 0.05 (=0.0167) | 0.35 (<0.0001) | 0.05 (=0.608) | 0.05 (=0.654) |
| Summer 2014 | 0.12 (<0.0001) | 0.26 (<0.0001) | -0.15 (=0.167) | 0.08 (=0.449) |
| Fall 2014 | -0.12 (=0.0001) | 0.33 (<0.0001) | 0.05 (=0.612) | 0.05 (=0.608) |
| Winter 2015 | 0.26 (<0.0001) | 0.57 (<0.0001) | 0.02 (=0.835) | 0.03 (=0.816) |
| Spring 2015 | 0.12 (<0.0001) | 0.48 (<0.0001) | 0.05 (=0.656) | 0.07 (=0.557) |






