# Peer review of "Impacts of Large-Scale Circulation on Urban Ambient Concentrations of Gaseous Elemental Mercury in New York, USA"

_Atmospheric Chemistry and Physics, 2017_

## Referee Comment (RC1) · Anonymous Referee #2 · 1 Jun 2017

This paper analyses seven years of GEM measurements in New York city. The authors observed large inter annual and seasonal variations in GEM concentrations and investigate the impact of mesoscale atmospheric transport patterns on Hg concentrations in NYC. The presented manuscript includes many interesting observations and conclusions but fails to present them in a consistent and concise way. I think that this paper would merit publication in ACP but only after significant improvement and a thorough correction.

P1 L45: 1 ng/m$^3$ = 112 ppqv (please add that ng/m$^3$ refer to standard conditions at 0°C and 1014hPa). I understand that it is a common convention to use mixing ratios

for air pollutants in the US. Nonetheless, I urge the authors to use ng/m$^3$ at standard conditions throughout the paper. This has been a common convention for Hg related studies even in US journals.

P4 L75pp: You should also mention that primary emissions due to more coal combustion is larger during winter time.

P6 L123: Please briefly describe the mentioned emission scenarios.

As you do not consider atmospheric chemistry, did you emit 100% of the Hg as GEM?

P7 L149-150: You state that GEM mixing rations were often larger during the warm seasons. Often is a very vague expression and I think that Figure 2 does not fully support this claim. Please give a quantitative measure e.g. the ratio of summer winter average mixing ratios for each year. Just looking at figure 2 I would conclude that this is true for 2014 and 2009 only.

P9 L181: Figure 3 is difficult to read as the different percentile values are not clearly distinguishable in the plot.

P9 L 190pp: You assume that the Hg emitted in the Eastern US is perfectly mixed within the PBL and ignore transport, chemistry, deposition as well as the temporal variability of emissions by simply dividing by 365 to get an daily change. I can fully understand that you start by estimating the maximal possible impact of emission changes on regional Hg mixing ratios but your calculation seems implausible. Your estimate for the Eastern US for 2008-2011 is -200 ppqv Hg (by the way please add signs to indicate increase or decrease before the numbers). This value is higher than the northern hemispheric average Hg mixing ratio of 168 ppqv (1.5 ng/m$^3$). This obviously makes no sense. For NYC you estimate even larger values with increases in the range of 6 ppqv per day.

In 2014 you observed mixing ratios 90 ppqv higher than in the previous year and state that: '90 ppqv throughout the seasonal averaged diurnal cycle, a factor of 15 larger than the effect of local emissions'. I can only guess that this is based on the calculation

90 ppav / 6 ppqv d-1 = 15. Which leads me to assume that you made a mistake with the units here. If you give the emissions in P9 L190 as kg/a and assume an average residence time for the emissions inside your domain of 1 day the resulting change in Hg mixing ratio would be 6ppqv and not 6ppqv d-1. This would at least lead to plausible values.

However, I still object this oversimplified approach. Given the fact that already in P9 L197 you state that this estimate is inconsistent with observations rises the question why you include this flawed approach in the first place. Why don't you use the results from the HYSPLIT simulation here?

P11 L222pp: Please elaborate whether the local wind directions are representative of the regional transport patterns. As your measurement station is located inside a major city local wind direction are not necessarily consistent. Please add more information ion the measurement site (e.g. height of measurement point compared to surrounding buildings) in P5 Section 2.

P11 L229pp: It would be highly interesting if you could calculate wind direction adjusted inter-annual changes in Hg mixing ratios.

P12 Figure 5: It would be of great value if you could add SO2 observations to this figure as they are an indicator of regional Hg sources (mainly coal combustion in the US).

P14 L301p: Please check grammar.

P17 L371pp: It is very interesting that GEM trends in all four wind quadrants were similar. Was this a local effect in NYC or did other Hg stations in the Eastern US see similar increases? Moreover, did you see a correlated increase in other pollutants (reference to the following section)?

P17 L380: Please clarify what emission increases could explain the much higher Hg mixing ratios in summer 2014. Otherwise, this sentence seems to be just a filler.

P18: Please do not swap between r and $r^2$.

Hg sources in the US should be much more linked to SO2 than CO. Thus, I would suggest to include the analysis for Hg/SO2 in Figure 9. Your finding that SO2 is only weakly correlated to Hg at the Bronx site is very interesting and needs some additional clarification. E.g. what sources were driving the SO2 emission reduction in the area? The Hg/SO2 ratio from local sources is also strongly influenced by the type of coal used for combustion. Maybe if you use the Hg/SO2 emission ratios for each you could normalize the observed ratio?

P18 L394pp: Please note that CO/Hg ratio is mostly indicative of Hg from biomass burning. In urban areas vehicles are a major CO source and will dominate the CO signal. Thus, I would argue that the Hg/CO ratio is mostly a sign of wind speed and buildup of regional air pollution in NYC and obviously will behave differently than at an urban stations.

Section 5: It is commendable that you include a modeling exercise into the paper. However, it seems just attached as a supplement in the end. You need to better describe the model setup and also openly discuss the weaknesses and shortcomings (e.g. long range transport, chemistry). And then include the model results already in the discussion of the other sections.

P19 Section 5: Please describe in more detail your model setup. In the introduction you mention 2 emission scenarios which are not further described anywhere. Moreover, it is important to know what you used as boundary conditions to estimate the background concentrations of Hg.

You estimate that 75% of the anthropogenically induces GEM in NYC originates from local sources (and with a larger domain this could even be enhanced). It seems that you underestimate the long range transport of Hg as your results do not seem consistent with findings from similar mode lstudies (e.g. Cohen et al., 2016. Modeling the atmospheric transport and deposition of mercury to the Great Lakes. Environmental Research)

P22 L 486: "... regional contributions to NYC ambient concentrations would be even more dominant." You need to clarify that this assumption neglects all long range transport of Hg.

---

## Referee Comment (RC2) · Anonymous Referee #3 · 4 Jun 2017

This is a very interesting manuscript that demonstrates the significant impact of large-scale circulation on trace gases in an urban area, especially GEM. This innovative work yields surprising results that show the importance of meteorology and how it can dominate over anthropogenic emission sources; a surprising result to some. Except for a few minor comments, the manuscript is ready for peer review.

p. 6, line 111 – what type of catalytic converter was used on the TEI42C? If it was the usual moly (molybdenum) converter from TEI, it actually measures NOy not NO2. The Moly converter efficiently (100%) converts NOy species to NO. A blue light converter will provide much more accurate measurements of NO2.

[Figure]

p. 9, lines 192 & 193 - these are surprisingly rapid increases in GEM.

p. 10, line 201 – this is an impressive increase in GEM. Was the calibration checked to ensure no issues with it that might have caused this change? I have never seen anything like this before. The reproducible sinusoidal fluctuations over several hours look to me like an instrument problem. What else could explain these? They are very pronounced and have characteristics of temperature fluctuations with where the instrument was housed. I would double-check these things.

p. 10, line 213 – what type of meteorological circulations would have caused this increase, and where did such a large source of GEM originate?

p. 19 – don't these significant decreases in SO2 and NO2 emissions additionally rule out these same sources as being important for GEM?
* * *

---

## Author Comment (AC1) · 25 Jul 2017

We are deeply grateful to the reviewer for their thorough, thoughtful, and constructive comments, which have helped to clarify and improve the manuscript. We have addressed each of the comments carefully as detailed below.

*This paper analyses seven years of GEM measurements in New York city. The authors observed large inter annual and seasonal variations in GEM concentrations and investigate the impact of mesoscale atmospheric transport patterns on Hg concentrations in NYC. The presented manuscript includes many interesting observations and conclusions but fails to present them in a consistent and concise way. I think that this paper would merit publication in ACP but only after significant improvement and a thorough correction.*

*P1 L45: 1 ng/m$_3$= 112 ppqv (please add that ng/m$_3$ refer to standard conditions at 0◦C and 1014hPa). I understand that it is a common convention to use mixing ratio for air pollutants in the US. Nonetheless, I urge the authors to use ng/m$_3$ at standard conditions throughout the paper. This has been a common convention for Hg related studies even in US journals.*

Per the reviewer's suggestion, "0°C and 1013.25 hPa" was added after "a standard atmosphere" (lines 43-44).

As the reviewer stated, mixing ratios units were conventionally used for trace gases in atmospheric sciences. For that very reason we made a point in using ppqv for TGM/GEM in the very first publication on atmospheric mercury from our group back in 2007 and ever since then we have been using mixing ratios in all our publications. Acknowledging the pervasive use of mass units ng m$^{-3}$ of TGM/GEM as well as for readers' convenience, in every one of our publication we have always noted the conversion between ng m$^{-3}$ and ppqv in a standard atmosphere as done in this manuscript. Over the past decade mixing ratios have come to be used more commonly for TGM/GEM in the literature. We hope that this explanation provides the context of our use of the mixing ratio units.

*P4 L75pp: You should also mention that primary emissions due to more coal combustion is larger during winter time.*

Per the reviewer's suggestion, "probably more coal combustion to produce energy for space heating" was added as the first factor contributing to wintertime annual maximum (line 75).

*P6 L123: Please briefly describe the mentioned emission scenarios.*
*As you do not consider atmospheric chemistry, did you emit 100% of the Hg as GEM?*

The two emission scenarios were originally described. The wording was revised to make the description stand out better: "One scenario included emissions from all the 522 counties, and the other excluded emission from the five boroughs in NYC" (lines 147 – 149).

The reviewer was right that all emissions of Hg provided in the EPA emission inventory were treated as emissions of GEM. This information was added now in the text, as "Note that the total emissions of Hg were treated as 100% GEM emissions". See line 144.

*P7 L149-150: You state that GEM mixing rations were often larger during the warm seasons. Often is a very vague expression and I think that Figure 2 does not fully support this claim. Please give a quantitative measure e.g. the ratio of summer winter average mixing ratios for each year. Just looking at figure 2 I would conclude that this is true for 2014 and 2009 only.*

We agree with the reviewer that our statement was vague and inaccurate. We added one more column and one more row to Table 1. The last row includes the statistical metrics of all data over the entire period of 2008 – 2015. Then we defined "larger GEM mixing ratios" to be values exceeding the $75^{th}$ percentile value (200 ppqv) of all data during the entire study. The last column "Frequency of higher values" includes the occurrences of larger GEM mixing ratios in percentage of the total number of data points in each warm (spring+ summer) and cold (fall and the following winter) season. The text was revised accordingly as follows (lines 173 – 181):

> "Annual cycles of 2009, 2010, and 2014 displayed larger GEM mixing ratios (>the $75^{th}$ percentile value of the entire study) during the warm seasons (summer-spring) than the cool seasons (fall-winter) (Fig. 2; Table 1), in agreement with previous urban site studies (Denis et al., 2006; Liu et al., 2007; Zhu et al., 2012; Zhang et al., 2013; Civerolo et al., 2014). The pattern of such annual cycles was evidenced in >20% (<10%) of the warm (cold) season in 2009 and 2010, and 67% (31%) of the warm (cold) season in 2014 experiencing larger GEM mixing ratios (Table 1). However, this pattern was not reproduced in 2011 and 2012, where the frequency of occurrence of larger GEM values was either comparable between the two seasons or slightly higher in the cold season."

*P9 L181: Figure 3 is difficult to read as the different percentile values are not clearly distinguishable in the plot.*

Figure 3 has been improved in two ways:

1. All the points below the $10^{th}$ and above the $90^{th}$ percentile values were replaced with plotting the $5^{th}$ and $95^{th}$ percentile values only, which cleared up the figure.

2. The median values were marked with thickened black lines, which helped readers to focus on the seasonal and interannual variability of the median value and made it easier to follow the ones above and below it.

*P9 L 190pp: You assume that the Hg emitted in the Eastern US is perfectly mixed within the PBL and ignore transport, chemistry, deposition as well as the temporal variability of emissions by simply dividing by 365 to get an daily change. I can fully understand that you start by estimating the maximal possible impact of emission changes on regional Hg mixing ratios but your calculation seems implausible. Your estimate for the Eastern US for 2008-2011 is -200 ppqv Hg (by the way please add signs to indicate increase or decrease before the numbers). This value is higher than the northern hemispheric average Hg mixing ratio of 168 ppqv (1.5 ng/m₃). This obviously makes no sense. For NYC you estimate even larger values with increases in the range of 6 ppqv per day. In 2014 you observed mixing ratios 90 ppqv higher than in the previous year and state that: '90 ppqv throughout the seasonal averaged diurnal cycle, a factor of 15 larger than the effect of local emissions'. I can only guess that this is based on the calculation 90 ppav / 6 ppqv d-1 = 15. Which leads me to assume that you made a mistake with the units here. If you give the emissions in P9 L190 as kg/a and assume an average residence time for the emissions inside your domain of 1 day the resulting change in Hg mixing ratio would be 6ppqv and not 6ppqv d-1. This would at least lead to plausible values.*
*However, I still object this oversimplified approach. Given the fact that already in P9 L197 you state that this estimate is inconsistent with observations rises the question why you include this flawed approach in the first place. Why don't you use the results from the HYSPLIT simulation here?*

These estimates were meant to provide some ballpark numbers for the effect of changes in anthropogenic emissions on ambient concentrations, although they were apparently very rough estimates. Our thoughts were that if the estimated contribution to NYC ambient GEM from changes in anthropogenic emissions was comparable to the observed changes in ambient GEM mixing ratios, then it could be hypothesized that the large interannual variation observed in GEM was caused by changes in Hg anthropogenic emissions. However, what our estimates suggested was that the potential contribution from changes in regional anthropogenic emissions were simply too small to matter, and that from NYC emissions was either too small or was inconsistent with the direction of change in ambient GEM levels. Therefore, the role of changes in anthropogenic emissions was ruled out from further analysis. The analysis was then focused on the impact of interannual variation in large-scale dynamics. This paper was organized in an incremental manner, from description of the general characteristics of the temporal variabilities in the dataset, to the key features in the interannual variation in annual cycles and its relationship to that in large-scale circulation, and finally to the demonstration of this hypothesized relationship using HYSPLIT simulations. Therefore, it seemed to be premature to introduce the model results here in our opinion.

The reviewer was correct about how we derived the 15 times larger increase in GEM in summer 2014 than the potential contribution from the 6 ppbv per day increase from NYC anthropogenic emission increases. The 90 ppqv increase was observed comparing the seasonal median values in summers 2011 and 2014. We Comparing the 10[th], 25[th], 75[th], and 90[th] percentile values between the two seasons, the increases were about 50, 70, 40, and 160 ppqv, respectively. It seems that significant increases occurred throughout the whole range of mixing ratios, and hence the increase in the median value was used for comparison with the contribution from emission increases. If the residence time of emitted GEM was 1 day, the total increase in ambient mixing ratio would be 6 ppbv from anthropogenic emission increases and would be even smaller

spreading throughout a diurnal cycle.  However, the diurnal cycle of seasonal median GEM in summer 2014 was elevated by 90 ppqv compared to the one in summer 2011.  Hence we suggested that the effect of the increase in anthropogenic emissions from 2011 to 2014 was negligible, and connected this point to the model simulation results in Section 5.

In this revised manuscript, we removed the summer 2014 data due to the sinusoidal fluctuations related to a temperature artifact that was identified recently by NYSDEC, who operates the site, and AMNet, who QA&QC the data.  The removal of the data did not change the fundamental findings of the study, as GEM levels at the site were increasing after the lowest point in winter 2011 through spring 2015 (Fig. 3).  Most importantly the increases in the seasons from winter 2014 through spring 2015 were consistent with the dynamical analysis.  In the revised version, we used spring 2014 data, instead of summer 2014, for the analysis.  We revised the text as follows to reflect this point:

> "Second, if the residence time of emitted GEM was 1 day, the total increase in ambient mixing ratio would be 6 ppqv d$^{-1}$ due to anthropogenic emission increases and would be even smaller spreading throughout the day, which was negligible compared to the ~60 ppqv increase observed in the spring 2011 seasonal average diurnal cycle compared to the spring 2014 one (Fig. 4).  The contribution from the NYC anthropogenic emissions to ambient GEM was further demonstrated using HYSPLIT simulations in Section 5." (lines 225 - 231)

Per the reviewer's suggestion, we revised the text to clearly indicate the direction of changes as follows:

> "Using an average PBL height of 1000 m over the Eastern US (the surface area for the Eastern US is 2.483x10$^{12}$ m$^2$), a decrease of 4115 kg from 2008 to 2011 emissions was converted to a total decrease of 200 ppqv over all days of the three years and averaged a decreasing rate of 0.2 ppqv d$^{-1}$, and an increase of 575 kg from 2011 to 2014 emissions was converted to a rate of ~0.03 ppqv d$^{-1}$". (lines 214 - 218).

As shown in the quoted statement above, this decrease of 200 ppqv was for ALL days of the three years between 2008 and 2011.  This was further broken down to a decrease of 0.2 ppqv per day.  It is negligible compared to the northern hemispheric average Hg mixing ratio of 168 ppqv (1.5 ng/m$^3$).

*P11 L222pp: Please elaborate whether the local wind directions are representative of the regional transport patterns. As your measurement station is located inside a major city local wind direction are not necessarily consistent. Please add more information on the measurement site (e.g. height of measurement point compared to surrounding buildings) in P5 Section 2.*

The monitoring site is located on the roof of the Pfizer Plant Resource Laboratory on the northern edge of the New York Botanical Garden in the north Bronx (40°52′05″N, 73°52′42″W; USEPA site ID 36-005-0133), and is also a National Atmospheric Deposition Network/National

Toxic Network (NADP/NTN) site. The height of the measurement point is about 9 m from ground surface, and winds arriving at the location are not significantly obstructed by immediate surroundings. The 100 ha New York Botanical Garden is in the midst of highways and mixed residential/commercial areas. New York City is a metropolitan area with >19 million people and the region has a long manufacturing, petrochemical, and industrial legacy that includes contamination from Hg and other toxic compounds. Continuous measurements of meteorological variables and various trace gas and toxic air pollutants are conducted at this site. Additional details on the site can be found on the NYS DEC website (http://www.dec.ny.gov/docs/air_pdf/2017 plan.pdf). This information was added in Section 2.1 (lines 94 – 108).

*P11 L229pp: It would be highly interesting if you could calculate wind direction adjusted inter-annual changes in Hg mixing ratios.*

We calculated wind direction adjusted GEM values, shown in dark grey lines and solid dots in Figure 5c. They tracked seasonal median GEM values very closely.

*P12 Figure 5: It would be of great value if you could add SO2 observations to this figure as they are an indicator of regional Hg sources (mainly coal combustion in the US).*

Per the reviewer's suggestion, Figure 5d, same as Figure 5c, was plotted for $SO_2$. The annual cycles of $SO_2$ were reproduced every year with the maximum in winter and minimum in summer. A precipitous dip in winter 2012 $SO_2$ occurred, more than a factor of 2 lower than those in winters of 2009 – 2011, and the winter $SO_2$ median values continued to decrease in the following years but to a much less degree. Consistent with GEM (Fig. 5c), the wind from the southwest directions (180°-270°) brought in air masses with the highest $SO_2$ levels in 2008 – 2011, especially in winter reaching 13-14 ppbv, followed by half the values in winters of 2012 – 2015. In contrast, the $SO_2$ mixing ratios were close in the other three wind quadrants. This is consistent with what GEM of the four wind quadrants suggested. One difference between the two is that air masses from the southeast appeared to be rich in GEM, too, whereas $SO_2$ in air from the southeast was low, close to that from the northwest and northeast. The only landmass southeast of the Bronx is Long Island, with limited major polluters. One confounding factor for this difference could be due to the ocean being a major of source of GEM. Moreover, the emission profiles for $SO_2$ and GEM in NYC and the Eastern US are different. Fuel combustion emissions comprised nearly 90% of the total emissions of $SO_2$ and about 70% for GEM.

Some of the discussion here was added in the text as follows:

> "This argument was strongly supported by $SO_2$ values in the four wind quadrants (Fig. 5d). Consistent with GEM (Fig. 5c), southwesterly (180°-270°) wind brought in air masses with the highest $SO_2$ levels in 2008 – 2011, especially in winter reaching 13-14 ppbv, followed by half the values in winters of 2012 – 2015. In contrast, the $SO_2$ mixing ratios were close in the other three wind quadrants. One difference between the two is that air masses from the southeast appeared to also be rich in GEM, whereas $SO_2$ in air from the southeast was low, close to that from the northwest and northeast. One

confounding factor for this difference could be due to the ocean being a major of source of GEM.  Moreover, the only landmass southeast of the Bronx is Long Island, with limited major polluters." (lines 263-272)

*P14 L301p: Please check grammar.*

Reworded as:

> "In the warm seasons, Bronx was on the periphery of the Bermuda High (Figs. 7c,f), where usually lower wind speed prevailed". (lines 332 – 333)

*P17 L371pp: It is very interesting that GEM trends in all four wind quadrants were similar. Was this a local effect in NYC or did other Hg stations in the Eastern US see similar increases? Moreover, did you see a correlated increase in other pollutants (reference to the following section)?*

Weiss-Penzias et al. (2016, STOTEN) presented a flat trend during 2008 – 2014 in regionally averaged GEM monthly median concentrations in three regions in the Eastern US, one of which includes our Bronx site (NY06), one site in Upstate New York (NY43), three sites in Maryland (MD08, MD98, and MD99), and one in Ohio (OH02).  For a forested site in Upstate New York (NY20), Zhou et al. (2017, ES&T) showed a decreasing trend in GEM over 1992 – 2014, and a visual check on the data over 2009 – 2014 suggested a slight decrease.  The Update New York sites NY43 (43.1463°N, 77.5482°W) and NY20 (43.97°N, 74.22°W) are farther up north and farther west inland, climatologically falling outside the influence of the subtropical high in spring, and farther out on the periphery of the subtropical high in summer compared to where Bronx is (40.8680°N, 73.8680°W). Our first thought is that they are in a different circulation regime from the one that influences Bronx.  The three sites in Maryland (MD08, MD98, and MD99) are climatologically under the influence of the subtropical high constantly in summer and frequently in spring, and one in Ohio (OH02) is on the western periphery of the Bermuda High ridge.  A visual examination of the data at OH02 suggested increases in GEM in 2014 and 2015.  The interannual variability in large-scale circulation on those four sites may be more manifested in the intensity of the subtropical high, less dramatic than that on Bronx.  It will take a systematic and in-depth study to characterize and quantify the influence of large-scale circulation on these sites.

Regarding the second question: we did discuss the interannual variations in CO, $SO_2$ and $NO_2$ in Section 4, and the corresponding figures were Figures S5 & S6.  What we discussed on CO was:

> "The high percentile values of CO at the Bronx site had been affected by anthropogenic emission reductions over the years, while the 10th and 25th percentile values (referred to as baseline CO in the literature) remained fairly constant in all seasons (Fig. S5).  Zhou et al. (2017) suggested that baseline CO in Northeastern US rural areas was controlled by a multitude of factors including global biomass emissions, large-scale circulation, and cyclone activity.  At the Bronx site, the low percentile value, close to regional baseline levels, was possibly determined by a range of factors, whose importance could have varied from year to year." (lines 432 - 439)

In Lines 440 – 469, we discussed significant reductions of $SO_2$ and $NO_x$ emissions from 2008, to 2011, and to 2014, and how $SO_2$ and $NO_2$ mixing ratios of different percentile values changed over the time. Briefly, emission reductions appeared to drive the declining trends in all percentile values. However, a closer look revealed that slight increases in the 5[th] and median values of $NO_2$ and $SO_2$ and a slowed decrease in the 95[th] percentile value of $SO_2$ (Figure 6S). Note that $NO_2$ and $SO_2$ are much shorter lived than GEM, and also the percentage changes of emissions of these two were much more significant than GEM. Hence the influence of physical processes was manifested in changes in atmospheric concentrations of $SO_2$ and $NO_2$ less conspicuously than in GEM. Moreover, we speculated that different magnitude (or different emission control regulations) of changes in emissions of GEM, $SO_2$, and $NO_x$ "possibly contributed to confounding the emission signature of GEM vs. $NO_x$ and altered that of GEM vs $SO_2$".

*P17 L380: Please clarify what emission increases could explain the much higher Hg mixing ratios in summer 2014. Otherwise, this sentence seems to be just a filler.*

That statement was poorly worded, and the intended point was not conveyed. It was revised as follows:

> "The fact that the GEM values in the two relatively more polluted quadrants exhibited excellent correlations with the overall values suggested that the trend in the ambient GEM mixing ratio was largely shaped by the variability of anthropogenic influence. Such influence may not necessarily be driven by changes in anthropogenic emissions but could be caused by strong ventilation or regional build-up of pollution as demonstrated in earlier discussions." (lines 404 – 409)

*P18: Please do not swap between r and r2.*

The use of r and $r^2$ in different places on this page was intended. At first, we examined the correlation between GEM and CO using r values and established statistically significant correlation between the two over 2008 – 2013 and no correlation over 2014-2015. Then we went on to use $r^2$ values, as illustrated in Figure 9, to characterize the covariance of the two tracers and further examined the variation in the slope values of the correlation over the time.

*Hg sources in the US should be much more linked to SO2 than CO. Thus, I would suggest to include the analysis for Hg/SO2 in Figure 9. Your finding that SO2 is only weakly correlated to Hg at the Bronx site is very interesting and needs some additional clarification. E.g. what sources were driving the SO2 emission reduction in the area? The Hg/SO2 ratio from local sources is also strongly influenced by the type of coal used for combustion. Maybe if you use the Hg/SO2 emission ratios for each you could normalize the observed ratio?*

Since $SO_2$ and GEM were correlated poorly, a regression analysis to obtain $r^2$ and slope values of such poor correlation became not very meaningful in our opinion. That was why $Hg/SO_2$ was not included in the analysis in Figure 9.

We did examine $SO_2$ emissions over the years, which were very different from those of GEM. Fuel combustion emissions comprised ~90% of total $SO_2$ emissions in NYC and in the eastern US (including states east of the Mississippi River). NYC $SO_2$ emissions decreased by 30% from 2008 to 2011 followed by a further decrease of 43% to 2014. Eastern US $SO_2$ emissions decreased by 48% from 2008 to 2011 furthered by another 29% decrease in 2014. Specifically, NYC launched a Clean Heat program in winter 2008 – 2009 resulting in a 69% decrease in $SO_2$ concentrations averaged over the city-wide street-level monitoring sites in winter 2012 – 2013 (NYC Health, 2013). Our Figures 5d and S6 also showed significant decreases in $SO_2$ mixing ratios at the Bronx site. This suggested that ambient $SO_2$ mixing ratios at the Bronx site were largely driven by in $SO_2$ emission reductions. In comparison, changes in anthropogenic emissions of Hg were of different direction and magnitude. As we stated in the manuscript, total Hg anthropogenic emissions in NYC were increased by 16% from 2008 to 2011, mainly in miscellaneous non-industrial NEC and waste disposal emissions, and further increased by 37% from 2011 to 2014 primarily in fuel combustion. Emissions of Hg in the Eastern U.S. decreased by 13% from 2008 to 2011 and increased by 2% from 2011 to 2014. Therefore, we speculated that in addition to very different lifetimes of $SO_2$ and GEM, such different changes in emissions possibly contributed to confounding the emission signature of GEM vs $SO_2$. Moreover, we examined the effect of localized emissions using the correlation between GEM exceeding seasonal 95[th] percentile values and corresponding $SO_2/NO_2$, and found little correlation (Table 2). This was an indication that the ratios of $Hg/SO_2$ or $Hg/NO_2$ should not be used for local emissions as many previous studies showed. The segment of lines 440 - 469 included the discussion here. Some of the information above was added into the text to improve the clarity of discussion.

*P18 L394pp: Please note that CO/Hg ratio is mostly indicative of Hg from biomass burning. In urban areas vehicles are a major CO source and will dominate the CO signal. Thus, I would argue that the Hg/CO ratio is mostly a sign of wind speed and buildup of regional air pollution in NYC and obviously will behave differently than at an urban stations.*

We completely agree with the reviewer that in urban areas mobile sources are a major source of CO. That's why the very fact of significant correlation between Hg and CO over 2008 – 2013 was a strong indication of regional influence on the Bronx site, and this is consistent with the findings from our analysis of the impact of large-scale circulation. This is exactly the same as the reviewer's argument here that "the Hg/CO ratio is mostly a sign of wind speed and buildup of regional air pollution" making this urban site quite different from many in the literature. This point was first conveyed in lines 413 – 418 and was further reinforced in lines 425 – 430.

*Section 5: It is commendable that you include a modeling exercise into the paper. However, it seems just attached as a supplement in the end. You need to better describe the model setup and also openly discuss the weaknesses and shortcomings (e.g. long range transport,*

*chemistry). And then include the model results already in the discussion of the other sections.*
*P19 Section 5: Please describe in more detail your model setup. In the introduction you mention*
*2 emission scenarios which are not further described anywhere. Moreover, it is important to*
*know what you used as boundary conditions to estimate the background concentrations of Hg.*
*You estimate that 75% of the anthropogenically induces GEM in NYC originates from local*
*sources (and with a larger domain this could even be enhanced). It seems that you underestimate*
*the long range transport of Hg as your results do not seem consistent with findings from similar*
*model studies (e.g. Cohen et al., 2016. Modeling the atmospheric transport and deposition of*
*mercury to the Great Lakes. Environmental Research*
*P22 L 486: ". . . regional contributions to NYC ambient concentrations would be even more*
*dominant." You need to clarify that this assumption neglects all long range transport of Hg.*

The model set-up, input data, and sensitivity scenarios were described in Section 2, lines 131 –
152. Per the reviewer's suggestion, more detailed were added to reflect the discussion below. See
lines 475 – 480, 524 – 533.

What the reviewer suggested is one way to apply the model results.  Our logic for the set-up of
the paper was first characterizing the general features in the temporal variation in GEM at the
site (Section 3.1) followed by focused questions addressed using a data analysis approach
(Sections 3.2, 3.3, and 3.4). A key hypothesis from the analysis was that this urban site in Bronx
was dominated by regional influence over local sources facilitated by large-scale circulation.
The logical follow-up, in our opinion, is then using model simulations to validate the hypothesis.
To do this, the National Oceanic and Atmospheric Administration (NOAA) Hybrid Single
Particle Lagrangian Integrated Trajectory model (HYSPLIT) dispersion version (Draxler and
Hess, 1997, 1998; Draxler, 1999; Stein et al., 2015) was employed. The dispersion of a pollutant
is calculated by assuming a fixed number of particles being advected about the model domain by
the mean wind field and spread by a turbulent component, and by assigning certain mass to a
particle, emissions are incorporated in the model (Stein et al., 2015).  In this study the model was
run in the forward mode for 120 hours.  Input data for the model are wind data only, usually
from reanalysis or meteorological model output.  No boundary conditions are needed, as the
model simulates the distribution of concentrations caused by the sources within the domain, not
considering influence outside of the domain.

Apparently, the HYSPLIT dispersion model accounts for only long range transport of a pollutant
from sources within the domain, without considering chemical transformation, gas-to-particle
partitioning, atmosphere-surface exchange of mercury, loss through deposition, and background
concentrations.  Understandably output of this model is not comparable to output of 3-D
chemical transport models, which supposedly encompass all chemical, physical, and dynamical
processes of the real atmosphere.  This is why our results could be different from the studies
using 3-D chemical transport models.  However, for a compound such as GEM with a lifetime of
6 – 12 months, dispersion model simulations would be adequate for providing ***relative***
contributions of regional and local sources to ambient concentrations at a location of interest in
continental midlatitudes. The goal of our modeling exercise was quantifying the relative
contributions (%) to pollutant concentrations in NYC from: 1) sources in NYC alone, and 2) all
the sources outside of NYC in the modeling domain. The contribution of NYC sources alone

represents local influence, whereas the contribution from sources outside of NYC represents regional influence.  To accomplish the goal, we designed two modeling scenarios:

1) Running the model with the mercury emission sources in all 522 counties within the domain;
2) Running the model with the mercury emission sources in all but the five boroughs in NYC.

Simulations of Scenario #2 quantifies the contribution of sources outside of NYC to Hg concentrations in NYC, and the difference in the concentrations in NYC between the two scenarios quantifies the contribution of sources outside of NYC to Hg concentrations in NYC. Note that the US EPA's National Emissions Inventory documents emissions on a county, annual basis.  The EPA emission amount for each county was broken down to an hourly rate by the annual emission divided by (365x24) for the dispersion model simulations.

Regarding Cohen et al.'s Environmental Research paper, "*Modeling the atmospheric transport and deposition of mercury to the Great Lakes*", which the reviewer mentioned, it seems to be published in 2004, not 2016.  If this was the paper the reviewer referred to, they estimated the contributions of distant sources to deposition over the Great Lakes, which is different from our estimates of contributions to ambient concentrations.  Also, there were at least three other major differences between theirs and our study. First, they used a much larger domain, including anthropogenic emissions from the entire US and Canada, whereas we used a much smaller domain in the Northeast US including 522 counties only.  Second, they used anthropogenic emissions from 1999, which are reportedly 192 tons per year from North America, of which 125 tons per year from the US (Butler et al., 2007, https://www.arl.noaa.gov/documents/reports/ MDN_report.pdf), compared to 52 tons per year from the US in 2011 (https://www.epa.gov/roe/). Third, the Great Lakes region is much farther in the North located in a different circulation regime from where Bronx is, which results in very different transport pathways. Moreover, gas-phase chemical transformation and deposition of GOM and PBM being considered in Cohen et al.'s study whereas not in our study, and the HYSPLIT dispersion version in Cohen et al. assumed dispersion of a puff, as opposed to the particle dispersion version used in our study. Therefore, it was difficult to compare our results with theirs.

---

## Author Comment (AC2) · 25 Jul 2017

We greatly appreciate the reviewer's perceptive and constructive comments. Not only did they help to clarify the presentation of our work but also prevent the use of erroneous data in analysis. We have addressed all of the comments carefully, as detailed below.

*This is a very interesting manuscript that demonstrates the significant impact of large-scale circulation on trace gases in an urban area, especially GEM. This innovative work yields surprising results that show the importance of meteorology and how it can dominate over anthropogenic emission sources; a surprising result to some. Except for a few minor comments, the manuscript is ready for peer review.*

*p. 6, line 111 – what type of catalytic converter was used on the TEI42C? If it was the usual moly (molybdenum) converter from TEI, it actually measures NOy not NO2. The Moly converter efficiently (100%) converts NOy species to NO. A blue light converter will provide much more accurate measurements of NO2*

The TEI42C uses a moly converter.  The converter is located in the analyzer, not mounted at 10 m high up on a tower, so technically it does not measure $NO_y$.  The moly convertor measures $NO_2$ plus some fraction of $NO_z$ ($=HNO_3+HONO+N_2O_5+$organic nitrates$+…$).  Dr. Jim Schwab of SUNY at Albany recently gave a presentation (http://www.nescaum.org/documents/nyc-metro-area-energy-air-quality-data-gaps-workshop) that suggests that in NYC (Queens College), that $NO_z$ can be on the order of 12% of the total $NO_y$, which means that $NO+NO_2$ is about 88% on average.  The reviewer is correct that the method is not specific to $NO_2$.  This is a shortcoming of the method.  For the purposes of NAAQS compliance in urban areas, it is not of a major concern to NYS DEC who is responsible for all the monitoring work.  However, for the purposes of detailed $NO_y$ budget analyses, it can make a difference.  In this study, we used $NO_2$ as a tracer for fuel combustion sources, so the $NO_2$ data, albeit imperfect, served the purpose to a large extent.  This information was added in the data and approach section for clarification. See lines 121-126

*p. 9, lines 192 & 193 - these are surprisingly rapid increases in GEM.*

*p. 10, line 201 – this is an impressive increase in GEM. Was the calibration checked to ensure no issues with it that might have caused this change? I have never seen anything like this before. The reproducible sinusoidal fluctuations over several hours look to me like an instrument problem. What else could explain these? They are very pronounced and have characteristics of temperature fluctuations with where the instrument was housed. I would double-check these things.*

After several discussions with the New York State Department of Conservation (NYSDEC), who operates the site, and the Atmospheric Mercury Network (AMNet), who QA&QC the data and was not aware of the problem previously, it was recommended that the summer 2014 data be removed because the 3-hour sinusoidal fluctuations appeared to be related to a temperature artifact.  However, the removal of the data did not change the fundamental findings of the study, as GEM levels at the site were increasing after the lowest point in winter 2011 through spring

2015 (Fig. 3). Most importantly the increases in the seasons from winter 2014 through spring 2015 were consistent with the dynamical analysis.

*p. 10, line 213 – what type of meteorological circulations would have caused this increase, and where did such a large source of GEM originate?*

In line 213, we hypothesized that the striking contrast of GEM levels in Bronx between winter 2010 and winter 2014 was predominantly caused by the interannual variation in large scale circulation over the region with a particularly strong and persistent influence of the subtropical high pressure system in 2014 – 2015. The northeastern U.S. is a region with relatively concentrated emissions of Hg among other pollutants. In general, one key factor for the Northeastern U.S. to stay relatively clean is strong ventilation, which is facilitated by dynamic systems such as cold frontal passages causing high wind often accompanied by precipitation. Meteorologically this region is under the influence of the Eastern U.S. coastal trough on the 500 hPa pressure level. The large variation in the position and intensity of this trough and naturally the subtropical high pressure system resulted in close association between frontal passages and the subtropical high. A frontal passage wipes polluted air off the continent and bring in relatively clean, whereas a strong influence of the subtropical high produces relatively stagnant conditions conducive to regional build-up of pollution. Our analysis suggested a persistent and strong influence of the subtropical high (i.e. Bermuda High) starting in winter 2014, lasting through the whole year of 2014 (now with summer removed), and extending to spring 2015, as supported by what was shown in Figures 5 – 8. Surface wind data showed in winter 2014 the least frequent and slowest northwesterly wind that could bring relatively clean Canadian air to the region and in summer 2014 the highest frequency of $<1 \text{m s}^{-1}$ wind (Fig. 5) indicating fairly calm conditions prevailing in the region during the entire season. Further examination of large scale circulation patterns suggested the East U.S. positioned near the axis to the front of the trough in winter 2014, backed by the most negative TAI and TII indices (Fig. 6d) as well as 500 hPa geopotential height and sea-level pressure maps (Figs. 6c,g).

*p. 19 – don't these significant decreases in $SO_2$ and $NO_2$ emissions additionally rule out these same sources as being important for GEM?*

We agree with the reviewer that significant decreases in $SO_2$ and $NO_2$ emissions would have indicated the diminishing importance of the common sources of $SO_2$, $NO_2$, and GEM. However, the US EPA emission inventories showed emissions of Hg changed in different sectors and in the opposite direction compared to changes in emissions of $SO_2$ and $NO_2$ from 2008 to 2011 and from 2011 to 2014. Specifically, NYC Hg anthropogenic emissions were increased by 16% from 2008 to 2011, mainly in miscellaneous non-industrial NEC and waste disposal emissions, and further increased by 37% from 2011 to 2014 primarily in fuel combustion. Eastern U.S. emissions of Hg decreased by 13% from 2008 to 2011 and increased by 2% from 2011 to 2014. In contrast, NYC $SO_2$ emissions decreased steadily by 30% from 2008 to 2011 followed by a further decrease of 43% to 2014, while Eastern U.S. emissions decreased by 48% from 2008 to 2011 and furthered by another 29% decrease in 2014. As for $NO_2$, fuel and mobile combustion emissions comprised >99.5% of the total $NO_x$ emissions in NYC and ~90% over the Eastern US. NYC $NO_x$ emissions changed insignificantly (1%) from 2008 to 2011 and decreased by 15% from 2011 to 2014, while Eastern US mobile and fuel combustion emissions were decreased by

16% and 33%, respectively, from 2008 to 2011, and further decreased by 13% and 9%, respectively, to 2014.  Therefore, changes in $SO_2$ and $NO_2$ emissions could not necessarily indicate the direction or magnitude of changes in Hg emissions.